# Methylglyoxal-derived hydroimidazolone, MG-H1, increases food intake by altering tyramine signaling via the GATA transcription factor ELT-3 in *Caenorhabditis elegans*

Muniesh Muthaiyan Shanmugam[1], Jyotiska Chaudhuri[1], Durai Sellegounder[1], Amit Kumar Sahu[1], Sanjib Guha[1], Manish Chamoli[1], Brian Hodge[1], Neelanjan Bose[1], Charis Amber[2], Dominique O Farrera[3], Gordon Lithgow[1], Richmond Sarpong[2], James J Galligan[3], Pankaj Kapahi[1,4]*

[1]The Buck Institute for Research on Aging, Novato, United States; [2]Department of Chemistry, University of California, Berkeley, Berkeley, United States; [3]Department of Pharmacology and Toxicology, College of Pharmacy, University of Arizona, Tucson, United States; [4]Department of Urology, University of California, San Francisco, San Francisco, United States

*For correspondence: Pkapahi@buckinstitute.org

**Abstract** The Maillard reaction, a chemical reaction between amino acids and sugars, is exploited to produce flavorful food ubiquitously, from the baking industry to our everyday lives. However, the Maillard reaction also occurs in all cells, from prokaryotes to eukaryotes, forming advanced glycation end-products (AGEs). AGEs are a heterogeneous group of compounds resulting from the irreversible reaction between biomolecules and α-dicarbonyls (α-DCs), including methylglyoxal (MGO), an unavoidable byproduct of anaerobic glycolysis and lipid peroxidation. We previously demonstrated that *Caenorhabditis elegans* mutants lacking the *glod-4* glyoxalase enzyme displayed enhanced accumulation of α-DCs, reduced lifespan, increased neuronal damage, and touch hypersensitivity. Here, we demonstrate that *glod-4* mutation increased food intake and identify that MGO-derived hydroimidazolone, MG-H1, is a mediator of the observed increase in food intake. RNAseq analysis in *glod-4* knockdown worms identified upregulation of several neurotransmitters and feeding genes. Suppressor screening of the overfeeding phenotype identified the *tdc-1*-tyramine-*tyra-2/ser-2* signaling as an essential pathway mediating AGE (MG-H1)-induced feeding in *glod-4* mutants. We also identified the *elt-3* GATA transcription factor as an essential upstream regulator for increased feeding upon accumulation of AGEs by partially controlling the expression of *tdc-1* gene. Furthermore, the lack of either *tdc-1* or *tyra-2/ser-2* receptors suppresses the reduced lifespan and rescues neuronal damage observed in *glod-4* mutants. Thus, in *C. elegans*, we identified an *elt-3* regulated tyramine-dependent pathway mediating the toxic effects of MG-H1 AGE. Understanding this signaling pathway may help understand hedonistic overfeeding behavior observed due to modern AGE-rich diets.

## Editor's evaluation

This work, examining how Advanced Glycation End-products (AGEs), commonly found in processed and other cooked foods, affect eating behavior and signaling in the nematode *C. elegans*, is in a fundamentally important area of research with clear translational potential for humans. The authors

present a combination of solid and compelling evidence to mechanistically study how AGEs affect eating behavior. The objectives of this study are not only to provide basic information relevant to phenomena that are likely to be conserved throughout the animal kingdom, but also to provide information that could be important in human health for the understanding of disorders caused by the consumption of processed foods.

## Introduction

Processed modern diets enriched with advanced glycation end-products (AGEs), formed by the Maillard reaction, are tempting to eat but at the same time deleterious for health (*Chaudhuri et al., 2018*; *Nowotny et al., 2018*; *Zhang et al., 2020*). In 1912, a French Chemist, L.C. Maillard, reported a reaction between glucose and glycine upon heating, resulting in the formation of brown pigments (*Maillard, 1912*). Later, the covalent bonds formed between carbohydrates and proteins during heating in a non-enzymatic browning reaction was named the Maillard reaction (*Jaeger et al., 2010*; *Liu et al., 2020*). Glycation is a part of the Maillard reaction, or browning of food, during cooking which enhances the taste, color, and aroma of the food to make it more palatable (*Maillard, 1912*; *Lima, 2013*). The Maillard reaction has revolutionized the food industry by playing an important role in food chemistry (*Machiels and Istasse, 2022*); however, this reaction also results in the formation of adverse AGEs as well as toxic byproducts including acrylamide (*Luca et al., 2010*; *Mottram et al., 2022*; *Stadler et al., 2002*).

In addition to food sources, AGEs are also endogenously produced in cells when α-dicarbonyl compounds (α-DCs) (such as glyoxal [GO], methylglyoxal [MGO], and 3-deoxyglucosone [3DG]) non-enzymatically react with biomolecules. α-DCs are unavoidable byproducts of cellular metabolisms, such as glycolysis and lipid peroxidation (*Figure 1*). AGEs include GO derivatives such as carboxymethyl

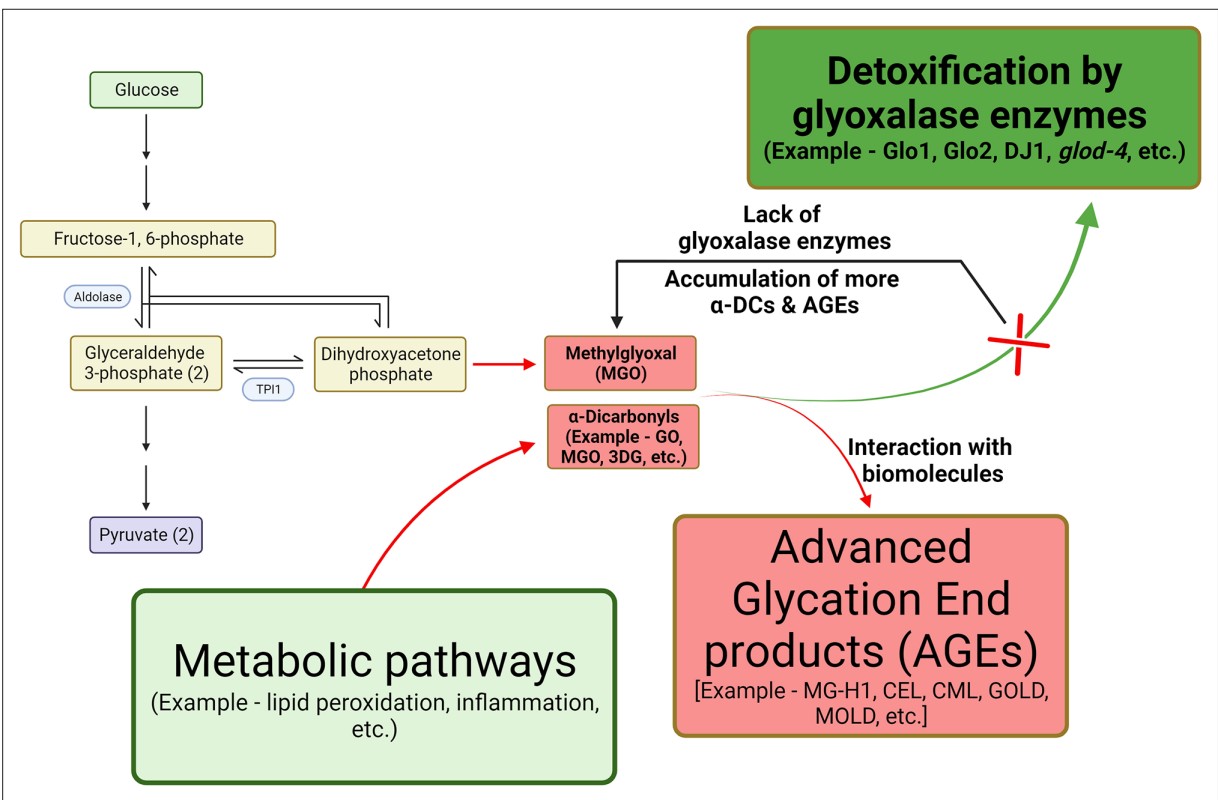

**Figure 1.** Graphical representation for the formation of α-dicarbonyls and advanced glycation end-products (AGEs). Dicarbonyls are highly reactive byproducts from metabolic pathways such as lipid peroxidation and glycolysis. In the above example, methylglyoxal (MGO) spontaneously forms from dihydroxyacetone phosphate which interacts with biomolecules resulting in the formation of AGEs. Toxic MGO is detoxified by glyoxalase enzymes to non-toxic lactate. One of the examples of glyoxalase enzyme in *C. elegans* is *glod-4*. Lack of glyoxalase enzyme leads to increased levels of MGO resulting in increased accumulation of AGEs.

lysine (CML) and glyoxal lysine dimer (GOLD). AGEs derived from MGO include hydroimidazolone (MG-H1), carboxyethyl lysine (CEL), and methylglyoxal lysine dimer (MOLD), and 3DG derivatives include 3-deoxyglucosone-derived imidazolium cross-link (DOGDIC), pyrraline, etc. (*Chaudhuri et al., 2018*; *Allaman et al., 2015*; *Chen et al., 2018*; *Vistoli et al., 2013*). The glyoxalase system utilizes enzymes Glo1 and Glo2 and reduced glutathione (GSH) to detoxify α-DCs stress, especially MGO to lactate (*Figure 1*), in cytosol and nucleus. Differential expression levels of glyoxalases are reported in various disease conditions such as diabetes, hypertension, neurodegenerative disorders, anxiety disorders, infertility, and cancer, suggesting their role in exacerbating their pathogenesis (*He et al., 2020*). Glo1 has been linked with several behavioral phenotypes, such as anxiety, depression, autism, and pain, among other mental illnesses (*Distler and Palmer, 2012*). Also, we have previously demonstrated increased neuronal damage in the *Caenorhabditis elegans glod-4* glyoxalase mutant model, which is shown to accumulate high levels of α-DCs and AGEs (*Figure 1* and *Figure 2—figure supplement 1H, I*); (*Chaudhuri et al., 2016*). AGEs accumulate in long-lived proteins, such as collagen (*Monnier et al., 1984*); furthermore, quantifying the glycated form of hemoglobin (HbA1c) is utilized as a biomarker in diabetes (*Rahbar et al., 1969*). Increased AGEs are associated with aging, obesity, diabetes, neurodegeneration, inflammation, cardiomyopathy, nephropathy, and other age-related diseases (*Chaudhuri et al., 2018*; *Ramasamy et al., 2005*; *Prasad et al., 2019*). Furthermore, neurodegenerative diseases have also demonstrated a strong correlation between increased levels of AGEs and pathogenesis.

Overconsumption of food and excessive availability of cheap, highly processed foods have contributed to the obesity pandemic. Obesity is a key risk factor for other diseases including diabetes, hypertension, cancers, cardiovascular, inflammatory, and neurodegenerative disorders, among other non-communicable chronic diseases (*Lee and Yau, 2020*; *Miller and Spencer, 2014*; *Wolin et al., 2010*; *Ellulu et al., 2017*; *Keramat et al., 2021*; *Uribarri et al., 2015*; *Ruhm, 2012*; *Hossain et al., 2007*). Thus, identifying signaling pathways that modulate increased feeding behavior is important to understand the underlying causes of obesity and identify novel therapeutics to overcome it. Here, we report that loss of the glyoxalase system or exogenously feeding MGO-derived AGEs increased feeding behavior in *C. elegans*. We also identified the mechanism for the observed phenotype and found that the MGO-derived AGE, MG-H1, acts via the *elt-3* GATA transcription factor (TF), to partially regulate the expression of *tdc-1* gene (tyramine decarboxylase – an enzyme that biosynthesis neurotransmitter tyramine), and tyramine receptors *(tyra-2* and *ser-2)* to mediate adverse effects of AGEs such as increased feeding, reduced lifespan, and neuronal damages. This study is the first to identify the signaling pathway mediated by specific AGEs molecules downstream of MGO (such as MG-H1) to enhance feeding and neurodegeneration. Our study emphasizes that AGEs accumulation is deleterious and enhances disease pathology in different conditions, including obesity and neurodegeneration. Hence, limiting AGEs accumulation is relevant to the global increase in obesity and other age-associated diseases.

## Results

### AGEs increase food intake and food-seeking behavior in *C. elegans*

Our initial observations revealed that *glod-4* glyoxalases enzyme mutants exhibit a significantly enhanced pharyngeal pumping than wildtype N2 animals (*Figure 2A*). This increase in pharyngeal pumping was consistent from day 1 (young adult, post-65 hr of timed egg laying) till day 3 of adulthood (*Figure 2A*). We performed a food clearance assay to validate whether increased pharyngeal pumping was accompanied by enhanced food intake (*Figure 2B* and *Figure 2—figure supplement 1A*). We found increased bacterial clearance after 72 hr in *glod-4* mutants.

Serotonin treatment increased bacterial clearance in both wildtype N2 worms and *glod-4* mutants (*Figure 2—figure supplement 1B, C*). Furthermore, worms lacking *tph-1* (tryptophan hydroxylase, an enzyme that catalyzes the formation of 5-hydroxy-tryptophan, precursor for serotonin) enzyme as well as *tph-1;glod-4* double mutants which lacks serotonin (*Dallière et al., 2017*; *Ji Ying Sze et al., 2000*) show decreased pumping compared to wildtype N2 worms (*Figure 2—figure supplement 1D*). However, *tph-1;glod-4* double mutants show significantly increased pumping compared to *tph-1* single mutants (*Figure 2—figure supplement 1D*). These data suggest that *glod-4* null mutation mediated increase in pharyngeal pumping is independent of the serotonin signaling (*Dallière et al.,*

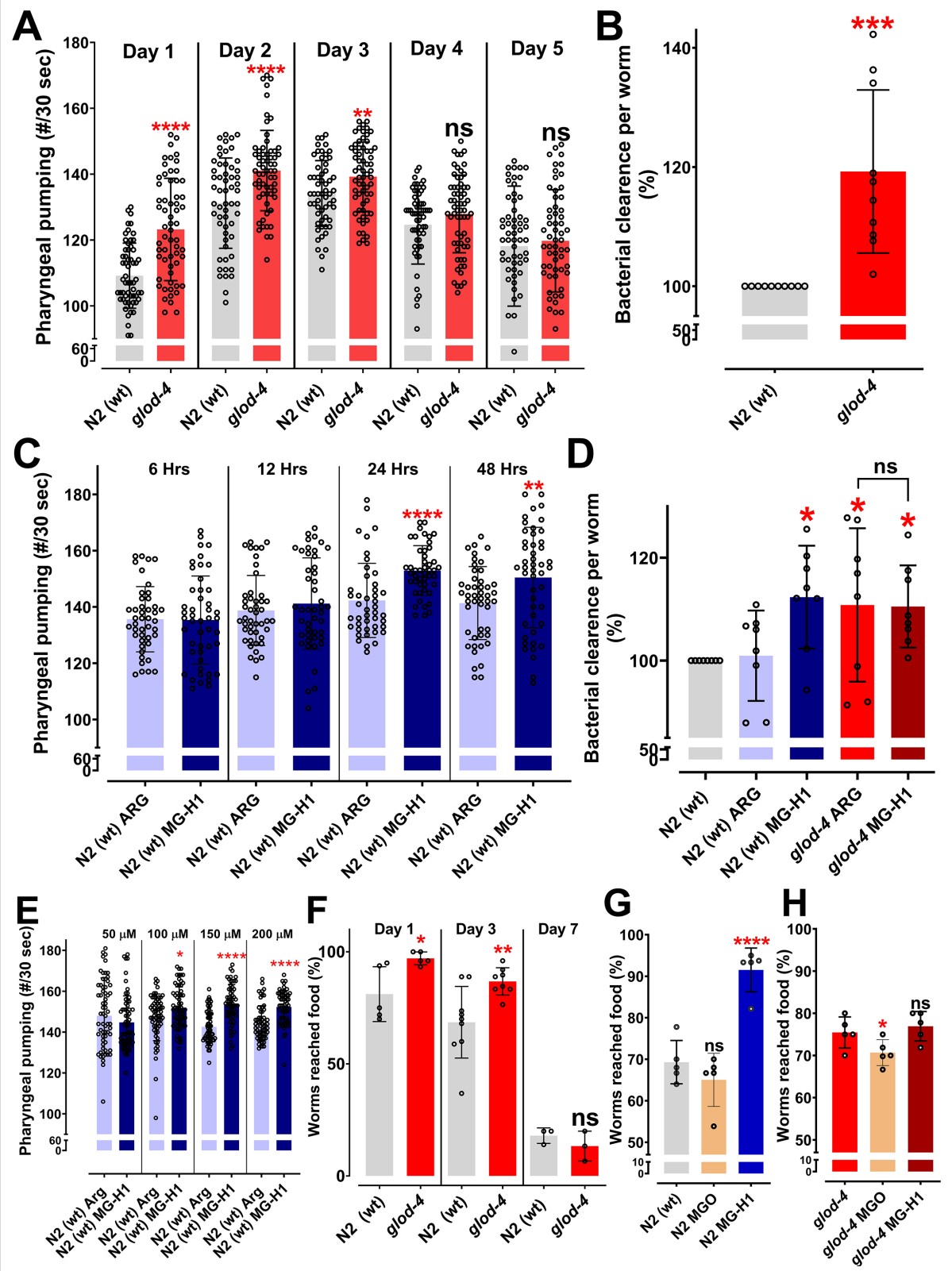

**Figure 2.** The glyoxalase mutant, *glod-4,* and methylglyoxal (MGO)-derived advanced glycation end-product (AGE), MG-H1, increases pharyngeal pumping and feeding in *C. elegans*. (**A**) Quantification of pharyngeal pumping (#/30 s) in N2 (wt) and *glod-4 (gk189)* mutant at different stages of adulthood. (**B**) Food clearance assay in N2 (wt) and *glod-4 (gk189)* mutant after 72 hr of feeding. (**C**) Quantification of pharyngeal pumping (#/30 s) in N2 (wt) after treatment, with either 150 µM of arginine (control) or MG-H1. (**D**) Food clearance assay in N2 (wt) and *glod-4 (gk189)* mutant worms after

*Figure 2 continued on next page*

*Figure 2 continued*

treatment for 72 hr with either 150 µM of arginine (control) or MG-H1. (**E**) Quantification of pharyngeal pumping with different concentrations of MG-H1. (**F**) Food racing assay in N2 (wt) and *glod-4 (gk189)* at different stages of adulthood toward OP50-1. (**G**) Food racing assay of N2 (wt) toward OP50-1 when combined with either MGO or MG-H1 (100 µM). (**H**) Food racing assay of *glod-4 (gk189)* mutants toward OP50-1 when combined with either MGO or MG-H1 (100 µM). Student's *t*-test for A, B, C, E, and F. One-way analysis of variance (ANOVA) with Fisher's LSD (Least Significant Difference) multiple comparison test for D, G, and H. The data points in the graphs represent the sample size (n). Comparison between two specific groups are indicated by lines above the bars; otherwise, the groups are compared with control group. *p < 0.05, **p < 0.01, ***p < 0.001, and ****p < 0.0001. Error bar ± standard deviation (SD).

The online version of this article includes the following figure supplement(s) for figure 2:

**Figure supplement 1.** The glyoxalase mutant, *glod-4,* and methylglyoxal (MGO)-derived advanced glycation end-product (AGE), MG-H1, increases pharyngeal pumping and feeding in *C. elegans.*

*2017*). These preliminary observations lead to the hypothesis that enhanced feeding in *glod-4* mutant worms is mediated by endogenous accumulation of AGEs characterized previously (*Chaudhuri et al., 2016*; *Golegaonkar et al., 2015*; *Morcos et al., 2008*). To this end, we explored AGEs such as MG-H1, CEL, CML, and F-ly as possible mediators of feeding and identified MG-H1 and CEL as potential MGO-derived AGEs to increase feeding in *C. elegans* (*Figure 2—figure supplement 1E*). Just feeding MGO was not sufficient to increase the pharyngeal pumping rate (*Figure 2—figure supplement 1E*). Time course analysis in wildtype N2 worms treated with MG-H1 showed that 24 hr of MG-H1 (150 µM) treatment was enough to increase pharyngeal pumping significantly (*Figure 2C*). A significant increase in bacterial clearance was observed after 72 hr (*Figure 2D*). Also, note that treatment of *glod-4* null mutants with MG-H1 did not further increase either the bacterial clearance or the pharyngeal pumping (*Figure 2D* and *Figure 2—figure supplement 1F*), suggesting that MG-H1 and *glod-4 null* mutation increases feeding by overlapping mechanism. In addition, we also demonstrated that MG-H1 regulates pharyngeal pumping rate in a dose-dependent manner (*Figure 2E*). Since MG-H1 is the product of arginine modification by MGO (see Materials and methods), we used arginine as a negative control for our MG-H1 treatment. We did not observe a significant difference between worms treated with arginine versus water versus phosphate-buffered saline (*Figure 2—figure supplement 1G*). Since MG-H1 induces increased pharyngeal pumping (*Figure 2C, D*), we validated an increase in MG-H1 in the *glod-4* null mutants compared to N2 wildtype using liquid chromatography–multiple reaction monitoring (LC–MRM) mass spectrometry (*Figure 2—figure supplement 1H, I*). In addition to food consumption, the *glod-4* mutant exhibited a significantly increased preference toward food source OP50-1 at days 1 and 3 of adulthood compared to wildtype N2 worms (*Figure 2F* and *Figure 2—figure supplement 1J*). Furthermore, we noticed that wildtype N2 worms preferred exogenous MG-H1 compared to MGO when provided with bacterial food source *Escherichia coli* OP50-1 (*Figure 2G*). We did not observe this phenotype in the *glod-4* mutant background (*Figure 2H*), suggesting MG-H1 in food makes it more appealing for control worms.

## Tyramine regulates MG-H1-mediated feeding behavior via G-protein-coupled receptors TYRA-2 and SER-2

Next, we sought to elucidate how MG-H1 increases the feeding behavior in worms. We performed an unbiased RNA sequencing approach to analyze the global transcriptome profile between control and *glod-4* knockdown worms (*Supplementary file 1*). Our RNAseq analysis identified a total of 20,277 genes, of which 5035 genes were significantly changed (2237 upregulated genes and 2798 downregulated genes) in *glod-4* RNAi knockdown worms compared to N2 wildtype. Gene set enrichment analysis showed that the functional category of genes regulating feeding behavior was significantly upregulated in *glod-4* knockdown worms (>twofold enrichment score) (*Figure 3—figure supplement 1*, red * marked GO category). This analysis supports our above observation that *glod-4* mutants have an altered feeding rate. Previous studies in *C. elegans* have documented the role of neurotransmitters in *C. elegans* feeding behavior (*Avery and Horvitz, 1990*; *Chase and Koelle, 2007*; *Trojanowski et al., 2016*) and we observed differential expression of 66 neurotransmitters and feeding genes (which comprises ~19% of the total feeding and neurotransmission-related genes in *C. elegans*) (*Figure 3A*).

We next tested the involvement of these neurotransmitter genes in regulating MG-H1-mediated feeding behavior and systematically analyzed (suppressor screen) MG-H1-induced feeding in the

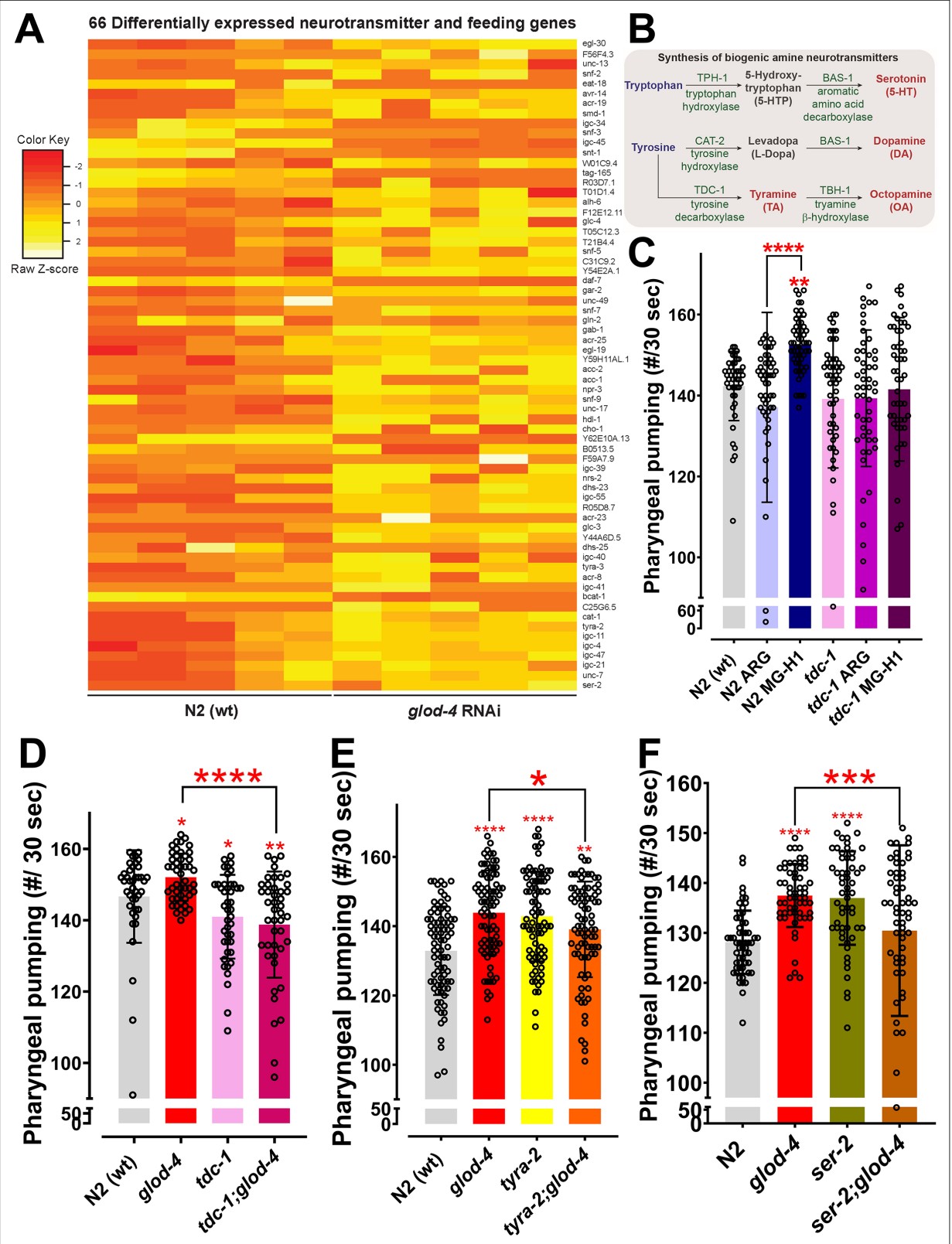

**Figure 3.** Role of *tdc-1* and tyramine receptors in mediating the MG-H1-induced feeding behavior. (**A**) Differential expression of 66 neurotransmitters and feeding genes in *glod-4* RNAi background. (**B**) The flowchart shows the pathway of biogenic amine synthesis, which functions as a neurotransmitter. (**C**) Quantification of pharyngeal pumping in N2 (wt) and *tdc-1* (n3419) mutant worms after 24 hr of treatment of MG-H1. (**D**) Quantification of pharyngeal pumping in N2 (wt), *tdc-1* (n3419), *glod-4* (gk189), and *tdc-1;glod-4* double mutants. (**E, F**) Quantification of pharyngeal pumping in N2 (wt), *tyra-2*

*Figure 3 continued on next page*

*Figure 3 continued*

(*tm1846*), *ser-2* (*ok2103*), *tyra-2;glod-4*, and *ser-2;glod-4* mutants. One-way analysis of variance (ANOVA) with Fisher's LSD multiple comparison test for C–F. The data points in the graphs represent the sample size (n). Comparison between two specific groups are indicated by lines above the bars; otherwise, the groups are compared with control group. $*p < 0.05$, $**p < 0.01$, $***p < 0.001$, and $****p < 0.0001$. Error bar ± standard deviation (SD).

The online version of this article includes the following figure supplement(s) for figure 3:

**Figure supplement 1.** Gene ontology analysis for upregulated genes in *glod-4* mutant worms.

**Figure supplement 2.** Role of *tdc-1* and tyramine receptors in mediating the MG-H1-induced feeding behavior.

background of genetic mutants limited in producing different biogenic amines and neurotransmitters in *C. elegans* (*Figure 3B* and *Figure 3—figure supplement 2A*). We found that mutation in *tdc-1*, the gene involved in synthesizing neurotransmitter tyramine, suppressed the enhanced feeding phenotype in MG-H1 treated animals (*Figure 3—figure supplement 2A*, indicated by a black arrow and *Figure 3C*). We also confirmed suppression of increased feeding rate in *tdc-1;glod-4* double mutant animals (utilizing two different *tdc-1* allelic mutants *n3419* and *n3420*) compared to *glod-4* single mutants (*Figure 3D* and *Figure 3—figure supplement 2B*). Next, we checked putative receptors for tyramine that could potentially mediate downstream signaling. Receptors for tyramine and octopamine are well-studied G-protein-coupled receptors (GPCRs) (*Gross et al., 2014*; *Pirri et al., 2009*). We screened seven GPCRs to identify the potential link in regulating tyramine-mediated increased feeding rate exhibited by *glod-4* mutant or MG-H 1-treated worms. Observed results showed a mutation in *ser-2* and *tyra-2* suppresses enhanced feeding in MG-H1 treated animals (*Figure 3—figure supplement 2C*, indicated by black arrows). A similar reversal of feeding phenotype was observed in *tyra-2;glod-4*, and *ser-2;glod-4* double mutant strains (*Figure 3E, F*). Our findings support the idea that MG-H1-induced overfeeding is mediated by tyramine signaling.

## GATA TF *elt-3* acts upstream of *tdc-1 to* regulate MG-H1-mediated feeding behavior

To check for putative TFs that could regulate the differentially expressed genes in *glod-4* knockdown worms (*Figure 3*), we performed a motif-enrichment analysis (based on available ChIP-Seq data) (*Figure 4A, B*). We chose the top five TFs (with a threshold of >18.75% target sequence match for TF-binding motif) for further screening. We knocked down each of the five TFs individually and checked for the suppression of MG-H1-induced feeding behavior (*Figure 4—figure supplement 1A*). Knocking down *pha-4* and *elt-3* suppressed the increase in pharyngeal pumping induced by MG-H1 treatment (*Figure 4—figure supplement 1A*, indicated by black arrows). The *pha-4* gene is crucial for pharynx development, and loss of *pha-4* results in a morphological defect of the pharynx (*Mango, 2007*; *Mango et al., 1994*), therefore, we followed the results from *elt-3* knockdown in *elt-3* mutants (*Figure 4C*).

Analysis of pharyngeal pumping in *elt-3;glod-4* double mutant showed that *elt-3* is essential to increase pharyngeal pumping observed in *glod-4* mutant worms (*Figure 4D*). Also, note that the knockdown of *elt-3* using RNAi feeding in *glod-4* single mutants suppressed the pumping (*Figure 4—figure supplement 1B*). To determine the role of *elt-3* in the tyramine signaling pathway, we performed a HOMER (Hypergeometric Optimization of Motif EnRichment) analysis and identified the binding site of *elt-3* on the *tdc-1* promoter, which suggested *elt-3* may potentially regulate *tdc-1* expression levels (*Figure 4—figure supplement 1C*). This was further validated by reduced expression of *tdc-1* mRNA levels in the *elt-3* mutant worms (*Figure 4E*). Next, to check if the *elt-3* expression is changed on exposure to MG-H1, we treated wildtype N2 worms with MG-H1 and quantified mRNA levels of *elt-3*. We observed a moderate but significant increase in the *elt-3* expression (*Figure 4F*). Although *tdc-1* and *tyra-2* did not change significantly, expression levels of other receptors, *tyra-3* and *ser-2*, increased significantly after MG-H1 exposure (*Figure 4F*). Note that *ser-2* is necessary to mediate the increased pharyngeal pumping (*Figure 3F*). Together, these experiments identified a key role for *elt-3* in tyramine-induced feeding increase in response to MG-H1.

## Tyramine signaling is necessary to increase feeding in *glod-4* mutants

To strengthen the role of tyramine in mediating increased feeding in *glod-4* KO worms, we treated mutants lacking tyramine signaling with exogenous tyramine. Tyramine has been demonstrated to

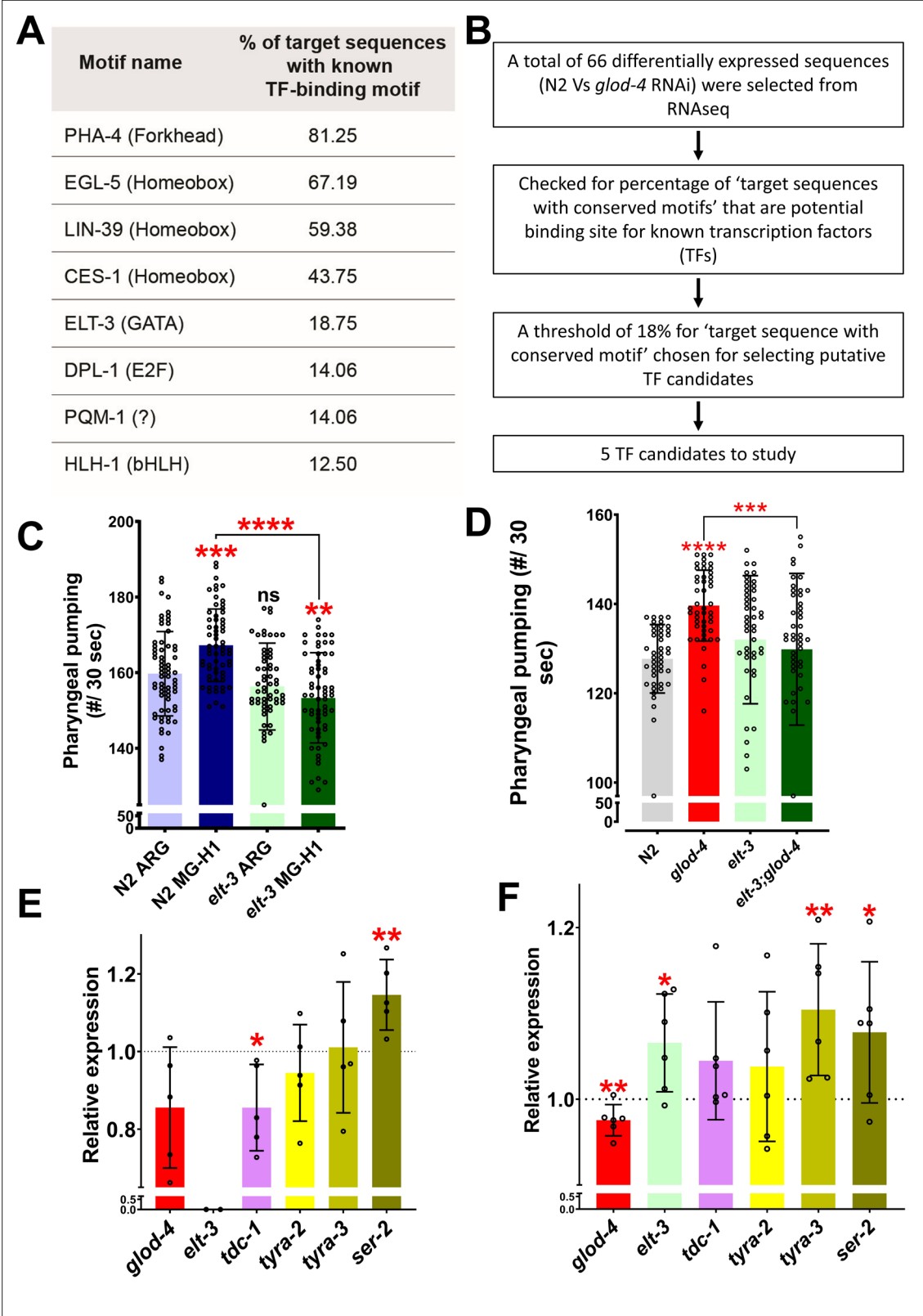

**Figure 4.** Role of *elt-3* transcription factor in regulating MG-H1-induced feeding in *C. elegans*. (**A**) List of transcription factors identified by motif analysis. (**B**) Flowchart demonstrating the method of identification of transcription factors. (**C**) Quantification of pharyngeal pumping after treatment with either arginine or MG-H1 in *elt-3 (gk121)* mutants. (**D**) Quantification of pharyngeal pumping in N2 (wt), *glod-4 (gk189)*, *elt-3 (gk121)*, and double mutant worms. (**E**) Quantification of tyramine pathway genes in *elt-3 (gk121)* mutant worms. (**F**) Quantification of *elt-3* and tyramine pathway genes expression in

*Figure 4 continued*

wildtype N2 (wt) worms after MG-H1 treatment. The horizontal dotted line indicates the normalized expression levels of genes in N2 (wt) and untreated control in E and F, respectively. One-way analysis of variance (ANOVA) with Fisher's LSD multiple comparison test for C, D. Student's *t*-test for E, F. The data points in the graphs represent the sample size (n) in C,D and number of biological repeats in E,F. Comparison between two specific groups are indicated by lines above the bars; otherwise, the groups are compared with control group. *p < 0.05, **p < 0.01, ***p < 0.001, and ****p < 0.0001. Error bars ± standard deviation (SD).

The online version of this article includes the following figure supplement(s) for figure 4:

**Figure supplement 1.** Role of *elt-3* transcription factor in regulating MG-H1-induced feeding in *C. elegans.*

decrease pharyngeal pumping in wildtype N2 worms (*Dallière et al., 2017*). Consistent with this finding, we found that exogenous treatment of tyramine significantly decreased pharyngeal pumping (*Figure 5A*). However, exogenous treatment of tyramine to rescue tyramine signaling in both *elt-3;glod-4* and *tdc-1;glod-4* double mutants shows a significant increase in pharyngeal pumping to the levels similar to that of *glod-4* single mutants (*Figure 5B, C*). Exogenous tyramine did not increase pumping in the double mutants lacking tyramine receptors such as *tyra-2;glod-4* and *ser-2;glod-4* (*Figure 5D, E*). These results strongly demonstrate that tyramine suppresses pharyngeal pumping in N2 wildtype worms; however, it increased the pumping in the *glod-4* mutant background. These data support the notion that tyramine signaling is necessary to mediate *glod-4* mutant-dependent increase in feeding behavior.

## Absence of tyramine rescues α-DCs and AGEs mediated pathogenic phenotypes

Accumulation of α-DCs in *glod-4* mutants results in pathogenic phenotypes, including neurodegeneration and shortening of lifespan (*Chaudhuri et al., 2016*). Here, chronic accumulation of MGO leads to the build-up of AGEs (*Figure 2—figure supplement 1I*), thereby increasing feeding in *glod-4* worms (*Figure 2*).

Furthermore, accumulation of MG-H1 significantly reduced the lifespan of N2 wildtype worms; however, it did not further exacerbate the damage in *glod-4* mutant worms (*Figure 6—figure supplement 1*). To test whether tyramine signaling is essential for mediating the pathogenic phenotypes such as neuronal damage and reduced lifespan in *glod-4* mutants, we compared the lifespan between wildtype N2 and *glod-4* worms in the genetic mutants that lack tyramine. The lifespan of *glod-4* was significantly increased upon inhibition of tyramine signaling in the *tdc-1;glod-4* double mutation (*Figure 6A*). Next, we tested if the absence of *tyra-2* and *ser-2* could also rescue the shortened lifespan of *glod-4* mutants. Lifespan increased significantly in the absence of either *tyra-2* or *ser-2* in double mutant animals (*Figure 6B, C*). In addition to rescuing lifespan and feeding rate, the lack of tyramine also resulted in the partial but significant rescue of neuronal damage in *glod-4* animals (*Figure 6D, E*).

## Discussion

Our observation that *glod-4* mutants run out of bacterial lawn faster than wildtype N2 animals during routine maintenance led to the elucidation of a novel signaling pathway that mediates AGE-induced feeding behavior in *C. elegans*. Glyoxalases are enzymes involved in the detoxification of α-DCs (*Figure 1*), and we have previously characterized *glod-4* mutant, which lacks one of the glyoxalase enzymes, to accumulate increased levels of α-DCs (*Chaudhuri et al., 2016*) and thereby AGEs, especially MG-H1 (*Figure 2—figure supplement 1H, I*). In this study, using genetic mutants, RNAi knockdown, synthesized AGEs, and functional genomics, we elucidate that AGEs (especially MG-H1) induce increased feeding through tyramine signaling regulated by GATA transcription factor ELT-3. The *glod-4* KO worms, with enhance AGEs accumulation, showed increased feeding, which led to the hypothesis that increased accumulation of AGEs is a potential stimulator of binge feeding. Thus, we studied changes in pumping rate by exogenous administration of MGO and AGEs. As previously reported by *Ravichandran et al., 2018*, MGO treatment did not change the pumping rate; however, MG-H1 and CEL increased the pumping rate in wildtype N2 worms (*Figure 2—figure supplement 1E*; *Ravichandran et al., 2018*). Furthermore, a recent study demonstrated that treatment with sugar-derived AGE-modified bovine serum albumin accelerated the pharyngeal pumping rate (*Papaevgeniou et al., 2019*). Our study demonstrates that either treatment with purified MG-H1 or endogenous

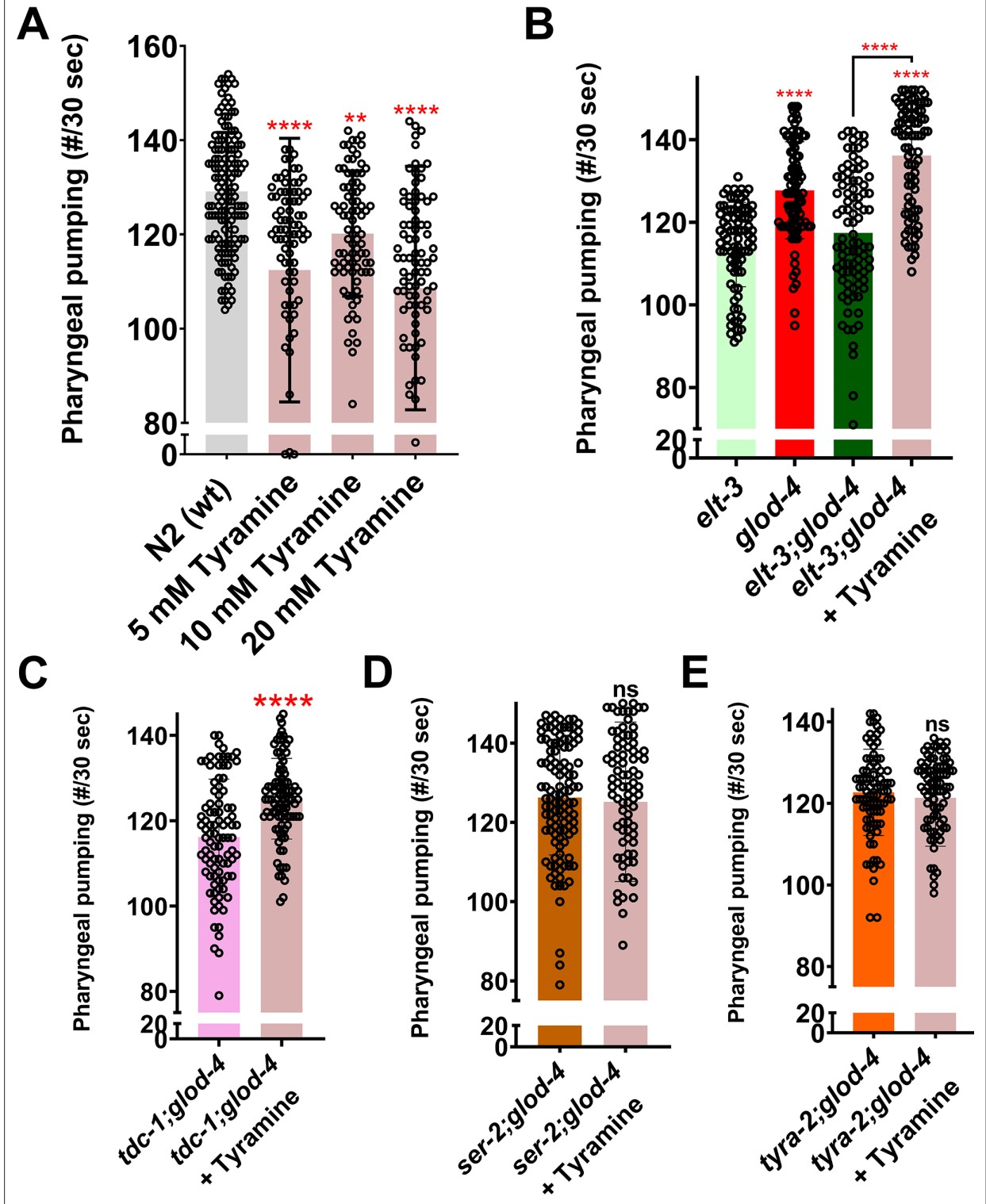

**Figure 5.** Exogenous tyramine rescues the suppressed pumping in double mutants. (**A**) Quantification of pharyngeal pumping in N2 wildtype worms treated with tyramine at various concentrations. (**B**) Quantification of pharyngeal pumping in *elt-3*, *glod-4*, *elt-3;glod-4* untreated and *elt-3;glod-4* double mutant treated with tyramine. (**C–E**) Quantification of pharyngeal pumping after treatment of double mutant worms (*tdc-1;glod-4*, *ser-2;glod-4*, *tyra-2;glod-4*) with exogenous tyramine. One-way analysis of variance (ANOVA) with Fisher's LSD multiple comparison test for A, B. Student's *t*-test for C–E. The data points in the graphs represent the sample size (n). Comparison between two specific groups are indicated by lines above the bars; otherwise, the groups are compared with control group. **p <0 .01, and ****p < 0.0001. Error bars ± standard deviation (SD).

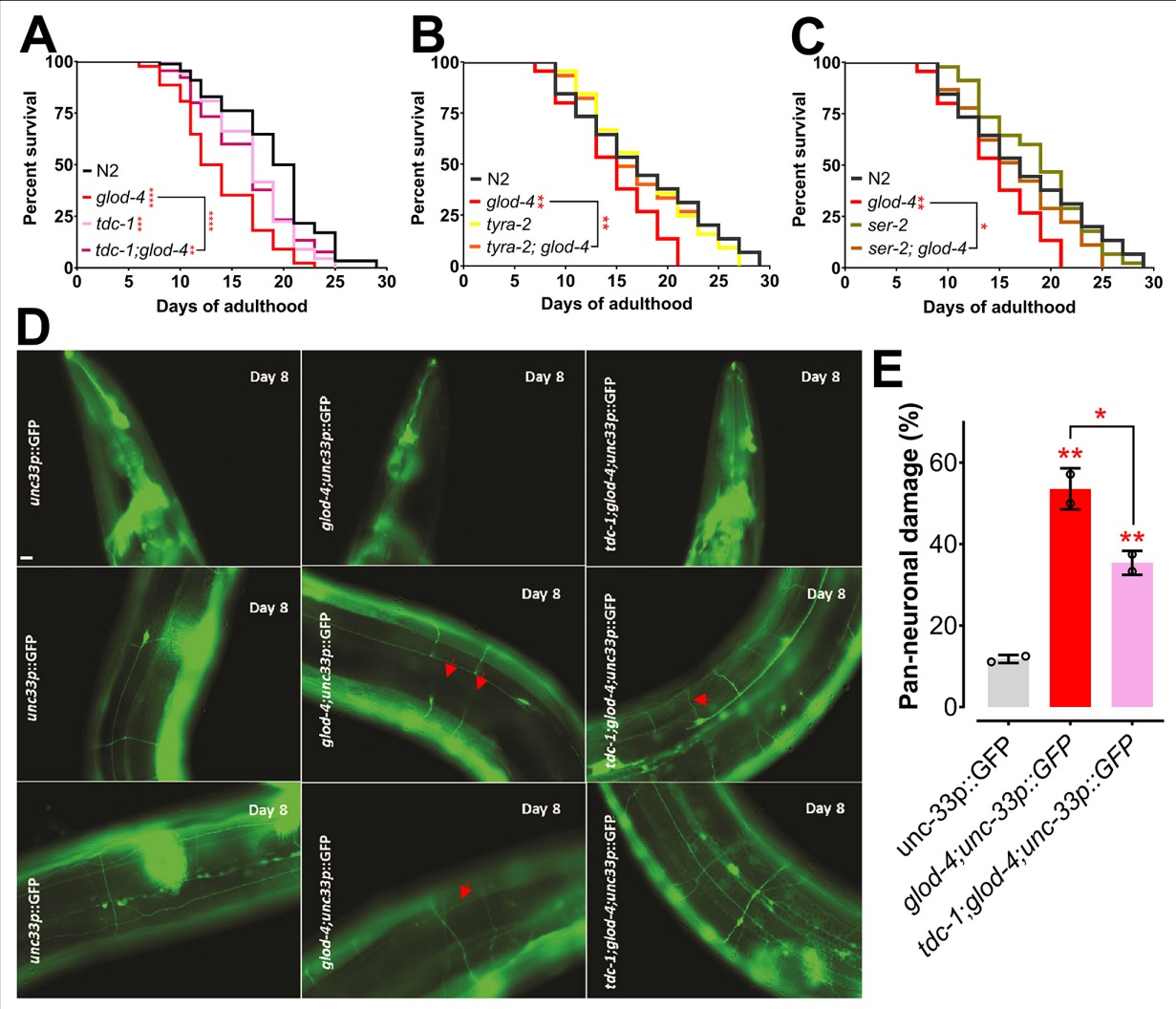

**Figure 6.** Suppression of *glod-4* phenotypes in *tdc-1;glod-4* double mutant. (**A**) Survival assay with N2 (wt), *tdc-1*, *glod-4*, and *tdc-1;glod-4* double mutants. (**B**) Survival assay with N2 (wt), *tyra-2*, *glod-4*, and *tyra-2;glod-4* double mutants. (**C**) Survival assay with N2 (wt), *ser-2*, *glod-4*, and *ser-2;glod-4* double mutants. (**D**) Image of worm neurons showing neuronal damage at day 8 of adulthood. Red arrows indicates damages. (**E**) Quantification of neuronal damage with pan-neuronal GFP marker in *glod-4* versus *tdc-1;glod-4* double mutants, 2 biological repeats. Scale bar – 10 μm. Log-rank (Mantel–Cox) test for survival assays. One-way analysis of variance (ANOVA) with Fisher's LSD multiple comparison test for E. Comparison between two specific groups are indicated by lines above the bars; otherwise, the groups are compared with control group. *p < 0.05, **p < 0.01, ***p < 0.001, and ****p < 0.0001. Error bar ± standard deviation (SD).

The online version of this article includes the following figure supplement(s) for figure 6:

**Figure supplement 1.** Survival analysis of N2 and *glod-4* mutant worms after MG-H1 treatment at 150 μM.

production and accumulation of MG-H1 via genetic mutation increases feeding and adversely affects lifespan. We also found that MG-H1-induced hyper-feeding is independent of serotonin-mediated hyper-feeding (*Dallière et al., 2017*) in *C. elegans* (*Figure 2—figure supplement 1B–D*).

Our detailed investigation of the time-dependent increase in pumping rate after MG-H1 treatment indicates a more robust and highly significant increase after 24 hr of treatment (*Figure 2C*). We also observed a MG-H1 dose-dependent increase in feeding rate (*Figure 2E*), which is based on the stronger significance (lower p-value) caused by reduced dispersion of the data at higher concentrations of MG-H1. Our analysis indicates that higher concentrations of MG-H1 can increase the pharyngeal pumping in almost all the treatment worms, thus predisposing the worms with lower pumping rates in the Gaussian distribution to higher pumping rates. It is well established that AGEs are formed during cooking, browning the food during dry heating, making the food more appetizing (*Vistoli*

*et al., 2013*). Furthermore, feeding is a multisensorial process regulated by several signaling pathways subjected to evolutionary adaptations (*Luca et al., 2010*). Thus, we wanted to analyze the changes in sensory behavior of *C. elegans* induced by either endogenous accumulation of AGEs or by exogenous administration of MG-H1 with the food. Since the *glod-4* mutant lacks glyoxalase system to detoxify MGO and leads to the accumulation of AGEs (*Figure 1*), the MG-H1-mediated signaling pathway can be responsible for the increased chemoattraction of *glod-4* mutant worms to food source OP50-1 (*Figure 2F*). It can be explained that including MG-H1 in bacterial lawn increased the chemoattraction of wildtype N2 worms toward food, resulting in increased attraction to palatable MG-H1-mixed bacterial food OP50-1 (*Figure 2G*). However, unlike wildtype N2 worms, exogenous MG-H1 treatment had no further increase in the feeding rate or chemoattraction of *glod-4* mutant worms (*Figure 2D, H* and *Figure 2—figure supplement 1F*), indicating the maximum sensory modulation attained by the endogenous accumulation of MG-H1 in the *glod-4* mutant. Although our screening identified CEL, a lysine-derived adduct of MGO, as another AGEs increasing the food intake, a detailed analysis is necessary to conclude the effect of CEL on feeding behavior (*Figure 2—figure supplement 1E*).

We utilized RNAseq data from *glod-4* knockdown worms to identify the novel signaling pathway that mediates AGE-induced feeding in *C. elegans*. Since *glod-4* knockdown data are enriched with several genes regulating the synthesis of neurotransmitters and feeding (*Figure 3A* and *Figure 3—figure supplement 1*), we performed suppression screening in mutant worms for genes involved in synthesizing biogenic amine neurotransmitters after MG-H1 treatment (*Figure 3B* and *Figure 3—figure supplement 2A, C*). Thus, our screen identified *tdc-1*, involved in tyramine biosynthesis, and tyramine receptors (*tyra-2* and *ser-2*) to mediate AGE-induced increased pharyngeal pumping (*Figure 3C–F* and *Figure 3—figure supplement 2*). Tryptophan and tyrosine are the substrates for synthesizing biogenic amines implicated in modulating various behaviors in *C. elegans* (*Chase and Koelle, 2007*; *Alkema et al., 2005*). Tyrosine to tyramine conversion in the presence of the enzyme tyrosine decarboxylase (TDC-1) followed by tyramine β-hydroxylase (TBH-1), is crucial for the synthesis of neurotransmitters tyramine and octopamine, respectively (*Alkema et al., 2005*; *Greer et al., 2008*). Previous studies have shown the role of tyramine and its receptor (*ser-2*) in regulating feeding and foraging behavior in *C. elegans* (*Dallière et al., 2017*; *Greer et al., 2008*; *Rex et al., 2004*; *Li et al., 2012*). Furthermore, the *tyra-2* receptor is expressed in MC (Marginal Cell) and NSM (NeuroSecretory Motor) pharyngeal neurons and is discussed to regulate pharyngeal pumping potentially (*Dallière et al., 2017*; *Rex et al., 2005*). Especially, tyramine has been shown to reduce pharyngeal pumping when applied exogenously to the worms (*Figure 5A*; *Dallière et al., 2017*). Supporting previous findings (*Dallière et al., 2017*), our observation shows increased pharyngeal pumping in *tyra-2* and *ser-2* single mutant worms (*Figure 3E, F*); at the same time, *tdc-1* single mutants did not increase pumping (*Figure 3D*). Converse to our observation of *tyra-2* and *ser-2* single mutants, *Greer et al., 2008* did not find any difference in the pumping rate of *tyra-2* and *ser-2* single mutants compared to wildtype N2 worms. However, the same study reported no changes in the pumping rate of the *tdc-1* single mutant, similar to our results (*Greer et al., 2008*), which is also demonstrated by *Li et al., 2012*. Interestingly, double mutants of either *tdc-1* or its receptors (*tyra-2*-partial suppression and *ser-2*) with *glod-4* mutant significantly suppress the increased pharyngeal pumping observed in either *glod-4* or *tyra-2* or *ser-2* single mutants (*Figure 3D–F*). It is to be noted that only two interneurons, namely RIM (Ring Interneuron M) and RIC (Ring Interneuron C) neurons, uv1 cells near vulva and gonadal sheath cells (*Alkema et al., 2005*) express the *tdc-1* gene, which is involved in the biosynthesis of tyramine; however, receptors of tyramine are expressed in distant tissues explaining an endocrine activity for tyramine neurotransmitter (*Dallière et al., 2017*) leading to the multi-pathway mode of action to exert differential response which should be elucidated in the future. Since *ser-3* mutant worms did not suppress the pumping (*Figure 3—figure supplement 2C*) and *ser-3* has been demonstrated to be a receptor for octopamine (*Dallière et al., 2017*), we conclude that octopamine is not responsible for mediating MG-H1-induced feeding in *C. elegans*.

Our suppressor screen for the upstream effector of the *tdc-1*-tyramine-*tyra-2/ser-2* pathway that mediates MG-H1-induced increased feeding identified the *elt-3* TF (*Figure 4C, D*). Thus, we examined whether *elt-3* TF regulates the *tdc-1*, *tyra-2*, or *ser-2*. Our analysis revealed that in *elt-3* mutant worms, the *tdc-1* gene is significantly reduced (*Figure 4E*), concluding that *elt-3* TF regulates tyramine biosynthesis. In favor of the data, HOMER analysis identified that the *tdc-1* gene is potentially regulated by *elt-3* TF (*Figure 4—figure supplement 1C*). Although *elt-3* TF is predominantly expressed

in hypodermal cells, its expression is also reported in the pharyngeal–intestinal valve, intestine, a few neurons (head neurons and mechanosensory PVD (Posterior Ventral process D) neuron), etc. ( Wormbase.org). In accordance with *elt-3* expression in PVD neurons and head neurons, *tyra-2* is also expressed in PVD neurons (*Rex et al., 2005*) and *tdc-1* in RIM and RIC head interneurons, respectively, suggesting a possible direct/partial regulation of *tdc-1* expression by *elt-3*. Also, *tyra-2* expression has been reported in pharyngeal MC neurons, which directly regulate pharyngeal pumping (*Rex et al., 2005*), suggesting direct endocrine action of tyramine. Similarly, *ser-2* is expressed in pharyngeal muscle segment cells (*Rex et al., 2004*; *Li et al., 2012*; *Tsalik et al., 2003*). Increased expression of *ser-2* in *elt-3* mutant worms can be inferred as a compensatory mechanism for reduced tyramine signaling by increasing the expression of the tyramine receptor. Also, *ser-2* expression is significantly increased in MG-H1 treated wildtype N2 worms (*Figure 4F*). Although the mechanism of MG-H1-induced expression of *ser-2* is unclear, it is evident that the *ser-2* genetic mutant can suppress the increased feeding in the *glod-4* mutant (double mutants) (*Figure 3F*), demonstrating an important role of the SER-2 receptor in mediating the MG-H1-induced feeding via tyramine. Furthermore, *elt-3* expression levels significantly increased after MG-H1 treatment. Altogether, our data strongly suggest the role of the *elt-3-tdc-1*-tyramine-*tyra-2/ser-2* pathway in mediating enhanced feeding. Finally, it is essential to note that the *ser-2* gene is upregulated in the *glod-4* knockdown RNAseq dataset, similar to significant upregulation after MG-H1 treatment, validating that MG-H1 is a critical player in mediating adverse phenotypes observed in *glod-4* mutant worms.

Exogenous tyramine suppressed pharyngeal pumping in N2 wildtype worms; however, tyramine rescued the pumping in double mutants (*elt-3;glod-4* and *tdc-1;glod-4*) to that of *glod-4* single mutants (*Figure 5*). Although our data strongly demonstrate the role of tyramine signaling in increasing feeding rate in *glod-4* mutant background, our study also identified a paradox in tyramine signaling to regulated pharyngeal pumping. From the literature (*Dallière et al., 2017*) as well as from our data, it is evident that tyramine suppresses pumping in N2 wildtype genetic background (*Figure 5A*). The mechanism behind this behavioral switch, that is, from suppressor of pharyngeal pumping in wild-type to a stimulator in *glod-4* mutant background, remains elusive. It can be speculated that MG-H1 can modify the tissue-specific expression of tyramine receptors (*Figure 4F*), resulting in an observed behavioral switch in response to tyramine signaling. This hypothesis can only be addressed by methodologies such as single-cell RNA sequencing and exploration of cellular signaling circuitry in further studies. Thus, our current understanding is that MG-H1 (either exogenous or endogenous accumulation in *glod-4* KO) modifies tyramine signaling by modulating genes in the tyramine pathway, leading to an increased feeding rate in *C. elegans*.

Previously, we have demonstrated reduced lifespan, hyperesthesia, and accelerated neurodegeneration-like phenotypes observed in diabetic conditions caused by excessive accumulation of α-DCs in *glod-4* mutant worms (*Chaudhuri et al., 2016*). Lack of dicarbonyl detoxification by glyoxalases enzyme in *glod-4* mutant worms (*Figure 1*) should result in the accumulation of AGEs (*Figure 2—figure supplement 1H, I*), which at sufficient concentration act as signaling molecules to modulate the feeding behavior (*Figure 2*) by causing differential gene expression (*Figure 3*). The increased amount of dicarbonyl stress, thereby AGEs, is observed in several systemic diseases such as obesity, diabetes, cardiovascular and neurodegenerative diseases, among other age-associated diseases (*Chaudhuri et al., 2018*). In diabetic patients, three times higher plasma levels of MGO have been reported and is a leading cause of neuropathic pain (*Bierhaus et al., 2012*; *Rabbani and Thornalley, 2014*; *Kold-Christensen et al., 2019*). Earlier reports in the literature show that the dicarbonyl levels correlate with diabetic complications. One of the major risk factors for diabetes is obesity (*Ismail et al., 2021*), which is caused by overfeeding. Thus, exploring the regulatory pathways of feeding is essential to understand and identify ways to modulate feeding behavior.

Here, we show that AGEs can modulate feeding behavior in evolutionary primitive model organisms, and it will be worth exploring this pathway in mammals. The TF *elt-3* belongs to the GATA TF family (*Gilleard et al., 1999*). *Shobatake et al., 2018* report that GATA 2 and 3 TFs induce the expression of appetite regulator genes such as POMC and CART (*Shobatake et al., 2018*). With the easy availability and unlimited access to modern-day processed food enriched in sugars and AGEs resulting in overeating, a significant cause of the obesity pandemic, it is necessary to explore signals regulating feeding. Importantly, our study shows exogenous treatment with MG-H1 increases feeding in worms (*Figure 2C, D*), indicating that a high AGEs diet in our day-to-day life can modulate feeding

behavior in humans. It is well known that food cooked by grilling, broiling, roasting, searing, and frying accelerates the formation of AGEs in food; thus, methods are explored to cook food with fewer AGEs accumulation (*Uribarri et al., 2010*). Furthermore, increased caloric intake and changes in eating habits have been reported in a behavioral variant of frontotemporal dementia (*Aiello et al., 2016*) and medication of antipsychotic drugs (*Perez-Gomez et al., 2018*).

Finally, we show that a lack of *tdc-1*-tyramine signaling rescues *glod-4* mutant phenotypes (lifespan and neuronal damage) (*Figure 6*). A strong association between worsening PD phenotypes with increased aggregation of α-synuclein and specific sites of increased glycation has been demonstrated in different genetic models with increased AGEs (*Vicente Miranda et al., 2017*). Thus, it will be interesting to investigate the role of the *tdc-1*-tyramine pathway in modulating pathways enhancing neurodegeneration and feeding. Recent research identified neurodegenerative diseases to be influenced by metabolism (*Muddapu et al., 2020*; *Procaccini et al., 2016*) and *glod-4* mutants demonstrate increased neuronal damage, decreased lifespan, and increased feeding. Thus, it is essential to investigate the balance in energy metabolism to identify critical pathways to modulate the outcome of neurodegenerative diseases.

## Materials and methods

### Strains

Strains were either obtained from *Caenorhabditis* Genetic Center (CGC), Minneapolis, USA or National Bioresource Project, Tokyo, Japan and the following strains were used: N2 (wt), VC343 *glod-4(gk189)*, VC143 *elt-3(gk121)*, MT13113 *tdc-1(n3419)*, MT10661 *tdc-1(n3420)*, FX1846 *tyra-2(tm1846)*, RB1690 *ser-2(ok2103)*, MT9455 *tbh-1(n3247)*, CB1112 *cat-2(e1112)*, MT15434 *tph-1(mg280)*, DA1814 *ser-1(ok345)*, RB1631 *ser-3(ok2007)*, RB745 *ser-4(ok512)*, VC125 *tyra-3(ok325)*, and OH438 otls117[*unc-33p::gfp + unc-4(+)*]. All the mutant strains were outcrossed at least three times or more with N2 wildtype. Mutant strains are crossed to get the double mutants *elt-3;glod-4*, *tdc-1(n3419);glod-4*, *tdc-1(n3420);glod-4*, *tph-1;glod-4*, *tyra-2;glod-4*, *ser-2;glod-4*, *glod-4;unc-33p::gfp*, and *tdc-1;glod-4;unc-33p::gfp*. RNAi clones were obtained from Ahringer's RNAi feeding library and the following were used: *pha-4*, *ces-1*, *elt-3*, *lin-39*, *egl-5*, and *tdc-1*.

### Growth and maintenance

Worms were cultured at 20°C for at least two generations on standard NGM (Nematode Growth Media) agar plates seeded with 5× *E. coli* OP50-1 bacterial strain (Broth culture of OP50-1 was cultured overnight at 37°C at 220 rpm), which was propagated at room temperature for 2 days. For feeding RNAi bacteria, synchronized L1 larvae were transferred to NGM plates containing 3 mM of isopropyl β-D-1-thiogalactopyranoside (IPTG; referred to as RNAi plates) seeded with 20× concentrated HT115 bacteria (cultured overnight at 37°C at 220 rpm), carrying the desired plasmid for RNAi of a specific gene or bacteria carrying empty vector pL4440 as control and allowed to grow on plates for 48 hr at 37°C. For drug assays, synchronized young adult worms (60–65 hr from egg laying) were transferred to NGM plates (with or without IPTG) with 20× HT115 RNAi bacteria or 5× OP50-1 bacteria, respectively, which are freshly overlayed by the desired drug (or vehicle control) that was air dried and diffused. Final drug concentrations were calculated considering the total media volume on the NGM plates.

Note: For *glod-4* mutant animals, we found that the pathogenic phenotypes discussed in this paper are contingent on strictly maintaining an ad libitum feeding regimen. Hence, care was taken not to allow the animals to starve by maintaining a low worm-to-bacteria ratio and transferring to fresh plates frequently (at least once every 2 days).

### Pharyngeal pumping assay

*C. elegans* pharyngeal pumping was measured using a Leica M165 FC stereomicroscope utilizing a modified previously established method (*Raizen et al., 2012*) on day 2 young adult worms (unless otherwise specified). Grinder movement in the terminal bulb was used as a read-out for the pumping rate phenotype. Pharyngeal pumping was recorded using a Leica M165 FC microscope; thus, obtained movies were played at ×0.25 times the original speed and a manual counter was used to count the number of pumps for 30 s. For quick pumping screening (pumping data in the figure supplements), the pumping rate was counted in real time for 30 s using a stopwatch and a manual counter focusing

the grinder using an Olympus SZ61 stereomicroscope. Ten to thirty animals were counted per biological repeat and two to three repeats were obtained for each experiment. The pumping data from all the repeats were combined for the presentation of data in the figures. At least one biological replicate was counted blind. Since pharyngeal pumping is very dynamic and changes with worms' development, we decided to record the minimum required treatment groups as possible to reduced variations caused by delayed time and worms' development. In case, when more treatment/genetic groups need to be compared the video recordings between different groups were staggered (recording of 10 worms per group followed by recording worms from other groups and repeating this cycle until 30 worms per group were recorded) to minimize the variations. To reduce the variations induced by fluctuations in the room temperature, 3 plates of worms were prepared for a single treatment group and only 10 worms were recorded per plate while other treatment plates were incubated at 20°C. Animals that did not pump during the recording time, worms that were stationary for prolonged amount of time and worms that are potentially injured with visible damages were eliminated from the analysis as well a few outliers were identified using the Gaussian distribution curve. Under exogenous drug treatment, animals were incubated in the drug at least 18–24 or until 48 hr before measurement of the pump rate. The drugs were overlaid on the NGM plate containing bacterial lawn and air dried before the addition of worms.

## Food clearance assay

Food clearance assay was performed following minor modifications to the established protocol by *Wu et al., 2019*. In brief, 20–25 age synchronized (L3–L4 stage) worms were washed twice in S basal then once with S complete medium and transferred to a 96-well plate containing 160 µl assay medium (S-complete medium, growth-arrested OP50-1 at final OD 0.8 (at 600 nm), antibiotics, FuDR and either 150 µM arginine or MG-H1 or 5 mM serotonin). Initial bacterial density was measured by obtaining OD at 600 nm. Following the indicated number of hours, bacterial density was measured at OD600 after a brief and gentle mixing using a multichannel pipet. For each experimental data point, at least six wells were measured (at least 120–150 worms in total), with the results shown being representative of at least two to three independent assays. The relative food intake was determined by the change in OD for each well, normalized to the number of worms. Under these conditions, ample OP50-1 was available for feeding throughout the analysis, and worms were maintained in the same wells for the entire duration of the experiment.

## Food race assay

The food race assay to evaluate *C. elegans* choice or attraction for a specific diet, a chemosensory behavior, was performed utilizing a previously established protocol (*Nyamsuren et al., 2007*). For this assay, synchronized adult worms (50 per race) were spotted on a 60-mm NGM agar plate, freshly seeded with *E. coli* OP50-1 (with or without drug) approximately 2 cm from the edge of the Petri plate. Adult animals were aliquoted on the plate diametrically opposite to the food source to estimate the percentage of worms that reached the food source within 30 min. An illustration of the food race assay has been provided (*Figure 2—figure supplement 1J*).

## Organic synthesis of AGEs

MG-H1 (**3**) was synthesized according to the literature procedure with a slight modification as follows: (L)-arginine (**1**) (6.07 g, 34.8 mmol, 1 equiv) was dissolved in 12 M HCl (50 ml). To this was added methylglycol dimethyl acetal (**2**) (4.53 g, 38.3 mmol, 1.1 equiv). It was then stirred at room temperature for 11 hr. At this time, the reaction mixture was diluted with water (200 ml) and concentrated *in vacuo.* The resulting dark-red solution was purified by $SiO_2$-gel column chromatography (4:2:1 ethyl acetate:methanol:acetic acid) to give MG-H1 (**3**) as a yellow solid (5.23 g, 22.9 mmol, 66%) (*Scheme 1*). The spectroscopic data obtained are consistent with those previously reported in the literature (*Hellwig et al., 2011*).

**Scheme 1.** Nδ-(5-hydro-5-methyl-4-imidazolon-2-yl)-ornithine (MG-H1) (3).

CML and CEL were synthesized according to the reported procedure (*Hellwig et al., 2011*) with a slight modification. To a 25-ml flask was added Nα-(*tert*-butoxycarbonyl)-L-lysine (**4**) (1.0 mmol, 1 equiv), palladium on carbon (10 wt% loading, 100 mg, 0.94 mmol), and distilled $H_2O$ (7 ml). To this was added glyoxylic acid (120 mg, 1.3 mmol, 1.3 equiv) for CML synthesis or pyruvic acid (115 mg, 1.3 mmol, 1.3 equiv) for CEL synthesis. 1 N NaOH$_{(aq)}$ was added dropwise to make pH of this solution 9. A balloon filled with hydrogen gas was attached, and the resulting solution was stirred at room temperature for 14 hr. At this time, the reaction mixture was filtered through a celite pad and the filtrate was concentrated *in vacuo*. Purification by SiO$_2$-gel column chromatography (1:2 ethyl acetate:methanol) yielded (**6a**) (260 mg, 0.85 mmol) or (**6b**) (263 mg, 0.83 mmol), respectively. To this was added 1 N HCl$_{(aq)}$ (3 ml), and it was then stirred at room temperature for 3 hr. The resulting solution was concentrated *in vacuo* to give CML (**7a**) (164 mg, 0.80 mmol, 80%) or CEL (**7b**) (172 mg, 0.79 mmol, 79%) (*Scheme 2*). The spectroscopic data obtained are consistent with those previously reported in the literature (*Hellwig et al., 2011*).

**Scheme 2.** Nε-carboxymethyl-lysine (CML) (7a) and Nε-(1-carboxyethyl)-lysine (CEL) (7b).

F-ly (**12**) was synthesized according to the literature procedure with a slight modification as follows: To a 200-ml round-bottomed flask was added Nα-(*tert*-butoxycarbonyl)-L-lysine (**4**) (510 mg, 2.8 mmol, 1 equiv), D-(+)-glucose (6.15 g, 30.0 mmol, 10 equiv), and MeOH (90 ml). The condenser was attached, and it was refluxed for 7 hr. After that, it was cooled to room temperature and concentrated *in vacuo*. The generated solid residue was purified by reversed-phase SiO$_2$-gel chromatography ($H_2O$ only) to provide the desired compound (**11**) in 53% yield (599 mg, 1.47 mmol). This compound was reacted with 1 N HCl$_{(aq)}$ (3.5 ml) at room temperature and stirred overnight. After the concentration *in vacuo*, F-ly (**12**) was obtained in 97% yield (438 mg, 1.42 mmol) (*Scheme 3*). The spectroscopic data obtained are consistent with those previously reported in the literature (*Thornalley et al., 1999*).

**Scheme 3.** Synthesis of F-ly (12).

## Preparation of samples and methodology for RNAseq

RNA preparation for RNAseq was performed using the Qiagen RNeasy Mini kit (Cat. No. 73404). Total RNA extraction was performed from day 1 adult animals (*n* = 30) picked and collected in 20 µl M9 buffer per condition. Five biological replicates were used for wildtype N2 and mutant animals. RNAseq on the extracted total RNA was executed at the University of Minnesota Genomics Core (UMGC) using their sequencing protocol for HiSeq 2500 High Output (HO) mode and 50 bp paired-end sequencing following Illumina Library Preparation. RNAseq coverage was ~22 million reads per sample to perform downstream bioinformatics analyses.

## Bioinformatic analysis

RNAseq global transcriptome data were subjected to Gene Ontology (GO) based functional classification using the Database for Annotation, Visualization and Integrated Discovery (DAVID) v.6.8. We employed the heatmap2 (Galaxy Version 3.0.1) function from R ggplot2 package to visualize the bioinformatics data.

HOMER analysis was used to identify the TFs for the 66 differentially expressed gene from *Figure 3A*. A threshold of 18% was used to select potential TFs for further screening. Please refer to the flowchart in *Figure 4B*.

## Reverse transcription polymerase chain reaction

Total RNA was extracted from nearly 100 µl of tightly packed age-synchronized adult worm pellet collected in 1 ml TRIzol reagent provided by Qiagen RNeasy Mini kit (Cat. No. 73404) following manufacturer's protocol. Subsequently, 1 µg total RNA was used as a template for cDNA synthesis. cDNA was synthesized using the iScriptTM cDNA synthesis kit (Bio-Rad, CA) following the manufacturer's protocol. q-PCR was carried out using the PCR Biosystems Sygreen Blue Mix Separat -ROX (Cat. No. 17-507DB) in a LightCycler 480 Real-Time PCR system (Roche Diagnostics Corp, IN). Quantification was performed using the comparative ΔΔCt method and normalization for internal reference was done using either *act-5 or pmp-2*. All assays were performed with 3 technical replicates followed by five to six biological replicates. Following are qPCR primers used: (1) *act-5* gene primers are 'TCCA ATCTATGAAGGATATGCCCTCCC and AAAGCTTCTCTTTGATGTCCCGGAC'. *pmp-2* primer pair is 'ATCTTTCAAAGCCAATCCTCGAC and GAGATAAGTCAGCCCAACTCC'. *glod-4*: TGTTCTGAATAT GAAAGTTCTTCGCCACG and GATGACGATTGCTCTATAATCATTACCCAACTC. *elt-3*: GCCGTTCA ATATTTTTGAATTGAACCTTTCAAACTT and TTTTTTTCATCGGCTTCGGCTCG. *tdc-1*: CGACGAGT TGTTCCTGCTATT and CGGATGTTGCCAATGAGTTATTC. *tyra-2*: GGAAGAGGAGGAAGAAGATA GCGAAAGTAG and ATCTCGCTTTTCATCCGAGTCTTCATC. *tyra-3*: CATCGATGGCCGCTTGGTC and CTTGTTCTCGGGTATTTGAGCGGT. And *ser-2*: GGAACAATTACGTACTTGGTAATTATTGCAAT GAC and ATATCGCCACCGCCAGATCG.

## Lifespan assay

Lifespan assays were performed in Thermo Scientific Precision incubators at 20°C. Timed egg laying was performed to obtain a synchronized animal population, which were either placed onto NGM plates seeded with 5× concentrated *E. coli* OP50-1. Post-L4 stage or young adult worms (60–65 hr from egg laying) were added to FuDR (5-fluoro-2 deoxyuridine) NGM plates to inhibit the development and growth of progeny. After 3 days, animals were transferred to a new 60 mm NGM seeded with OP50-1 and scored every other day thereafter. Forty-five to eighty animals were considered for each lifespan experiment, and two to three biological replicates were performed. Animal viability was assessed visually and with gentle prodding on the head. Animals were censored in the event of internal hatching of the larvae, body rupture, or crawling of larvae from the plates (*Chaudhuri et al., 2016*).

## Assay for assessing neuronal damage

Neuronal damage was assayed using a pan-neuronal GFP reporter strain under different conditions on day 8 of adulthood. Animals were paralyzed using freshly prepared 5 mM levamisole in M9 buffer and mounted on 2% agar pads under glass coverslips. Neuronal damage was visually inspected under an upright Olympus BX51 compound microscope coupled with a Hamatsu Ocra ER digital camera. Images were acquired under the ×40 objective. Neuronal deterioration was examined and characterized by loss of fluorescent intensity of nerve ring, abnormal branching of axon/dendrite, and thinning and fragmentation of axons and neuronal commissures (*Bijwadia et al., 2021*). Quantification and imaging of animals harboring damage were performed using the ImageJ software (http://imagej.nih.gov/ij/). To reduce experimental bias, this assay was performed genotype blind with two biological repeats.

## Mass spectrometry quantification of MG-H1

MG-H1 in worm homogenates were quantified using LC–MRM following 2,4,6-trinitrobenzene sulfonate derivatization as described in *Hashimoto et al., 2013*. An Agilent 1260 HPLC connected to a Sciex 5500 QQQ mass spectrometer was used. The LC conditions were modified as the follows: an Acquity UPLC BEH C18 (130 Å, 1.7 μm, 2.1 mm × 30 mm) column was used. The mobile phase (organic: methanol, aqueous: water containing 0.1% FA) was used with a linear gradient of 0–50% of the organic mobile phase over 1.5 min at 0.40 ml/min at 40°C. The gradient was increased to 100% organic at 1.6 min, held for 0.4 min, and equilibrated for an additional 0.4 min. The injections volume was 3 μl. MRM parameters were used as described in Hashimoto et al. with instrument-specific optimization (*Hashimoto et al., 2013*).

## Statistical analysis

All data analyses for lifespan, pharyngeal pumping assays, and gene expression were performed using GraphPad Prism (GraphPad Software, Inc, La Jolla, CA). Survival curves were plotted using the Kaplan–Meier method, and a comparison between the survival curves to measure significance (p values) was performed using log-rank (Mantel–Cox) test. Two groups were compared for significance using an unpaired Student's $t$-test. Multiple group comparison was performed by one-way analysis of variance with either Fisher's LSD or Dunnett's multiple comparisons test, or Sidak's multiple comparisons test was used to compare between specific groups. p values from the significance testing were designated as follows: $*p < 0.05$, $**p < 0.01$, $***p < 0.001$, and $****p < 0.0001$.

# Acknowledgements

This work was supported by grants from NIH (R01AG061165 and R01AG068288) to PK, the Larry L Hillblom Foundation (2021-A-007-FEL) to PK as well as R01DK133196 and R35GM137910 to JJG. We thank Professors Keith Blackwell, Suneil Koliwad, Malene Hansen, and the members of the Kapahi lab for their valuable suggestions. We thank Dr. Feimei Zhu for her guidance in troubleshooting the food clearance assay, Dr. Kiyomi Kaneshiro for her guidance in organizing the RNAseq data and Professor Jingru Sun as well as Phillip Wibisono for material support. We also thank the Buck Institute's morphology and imaging core for their valuable assistance.

# Additional information

### Competing interests

Pankaj Kapahi: Reviewing editor, eLife. The other authors declare that no competing interests exist.

### Funding

| Funder | Grant reference number | Author |
| --- | --- | --- |
| National Institutes of Health | R01AG061165 | Pankaj Kapahi |

| Funder | Grant reference number | Author |
|---|---|---|
| National Institutes of Health | R01AG068288 | Pankaj Kapahi |
| Larry L. Hillblom Foundation | 2021-A-007-FEL | Pankaj Kapahi |
| National Institutes of Health | R01DK133196 | James J Galligan |
| National Institutes of Health | R35GM137910 | James J Galligan |

The funders had no role in study design, data collection, and interpretation, or the decision to submit the work for publication.

### Author contributions

Muniesh Muthaiyan Shanmugam, Conceptualization, Data curation, Formal analysis, Validation, Investigation, Visualization, Methodology, Writing – original draft, Writing – review and editing; Jyotiska Chaudhuri, Conceptualization, Data curation, Formal analysis, Methodology, Writing – original draft; Durai Sellegounder, Conceptualization, Data curation, Formal analysis, Writing – review and editing; Amit Kumar Sahu, Manish Chamoli, Data curation, Formal analysis, Writing – review and editing; Sanjib Guha, Conceptualization, Data curation, Formal analysis, Writing – original draft; Brian Hodge, Formal analysis; Neelanjan Bose, Data curation, Formal analysis, Methodology; Charis Amber, Richmond Sarpong, Formal analysis, Methodology; Dominique O Farrera, James J Galligan, Data curation, Formal analysis; Gordon Lithgow, Supervision, Funding acquisition; Pankaj Kapahi, Conceptualization, Resources, Supervision, Funding acquisition, Validation, Investigation, Visualization, Methodology, Writing – original draft, Project administration, Writing – review and editing

### Author ORCIDs

Muniesh Muthaiyan Shanmugam ⓘ http://orcid.org/0000-0001-8018-2032
Durai Sellegounder ⓘ http://orcid.org/0000-0002-0776-0307
Amit Kumar Sahu ⓘ http://orcid.org/0000-0003-0063-5447
Manish Chamoli ⓘ http://orcid.org/0000-0003-0339-7894
Gordon Lithgow ⓘ http://orcid.org/0000-0002-8953-3043
James J Galligan ⓘ http://orcid.org/0000-0002-5612-0680
Pankaj Kapahi ⓘ https://orcid.org/0000-0002-5629-4947

### Decision letter and Author response

Decision letter https://doi.org/10.7554/eLife.82446.sa1
Author response https://doi.org/10.7554/eLife.82446.sa2

# Additional files

### Supplementary files

• Supplementary file 1. A list of genes identified in RNAseq between N2 wildtype and *glod-4* knockdown along with the data of fold change and significance.

• MDAR checklist

### Data availability

All data generated during this study are included in the manuscript. The RNAseq data is included as supplementary file.

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
