## [Editor Report]

This work, examining how Advanced Glycation End-products (AGEs), commonly found in processed and other cooked foods, affect eating behavior and signaling in the nematode *C. elegans*, is in a fundamentally important area of research with clear translational potential for humans. The authors present a combination of solid and compelling evidence to mechanistically study how AGEs affect eating behavior. The objectives of this study are not only to provide basic information relevant to phenomena that are likely to be conserved throughout the animal kingdom, but also to provide information that could be important in human health for the understanding of disorders caused by the consumption of processed foods.

---

## [Decision Letter]

**Decision letter after peer review:**

Thank you for submitting your article "Methylglyoxal-derived hydroimidazolone, MG-H1, increases food intake by altering tyramine signaling via the GATA transcription factor ELT-3 in *Caenorhabditis elegans*" for consideration by *eLife*. Your article has been reviewed by 3 peer reviewers, and the evaluation has been overseen by a Reviewing Editor and Jessica Tyler as the Senior Editor. The reviewers have opted to remain anonymous.

Essential revisions:

1) As you will see from the reviews attached below, all reviewers found this manuscript interesting and were enthusiastic about the central idea of the work. However, all three reviewers also agreed that there were mechanistic deficiencies in the manuscript that make it unsuitable for publication in *eLife* without additional data. In particular the reviewers agreed that mechanistic data about how MGH1 acts on the tyraminergic pathway to potentiate food intake were lacking in several places and weak in others (e.g., the elt-3 and tdc-1 data), and require strengthening to better support the proposed model. Specific rescues (elt-3 and tdc-1), functional experiments, and manipulation of tyramine-producing cell activity are all important approaches to obtain mechanistic information that will help identify how some of these players interact with each other (e.g., the cells and circuits that govern the processes).

2) Additional experiments, such as bacterial clearance (reviewer 1, comment 4), and something to rectify reviewer 3, comment 2, should also be completed.

3) Additionally, please note that both reviewer 1 and reviewer 3 point out several instances in the writing and/or figures that require clarification/revision, although these revisions may or may not require data depending on the response (e.g., pumping controls, backcrossing of worms). All reviewer comments are included below for individual responses.

*Reviewer #1 (Recommendations for the authors):*

These are the points that I feel need to be addressed:

1) The authors claim that MGH-1 modulates (positively) the tyraminergic signal through the activation of the TF elt-3. This is based on the fact that mutations in both tdc-1 and elt-3 suppress the exacerbated pharyngeal pumping of glod-4 mutants (Figure 2E and 3D), that elt-3 expression is increased in MGH1-treated worms (Figure 3F) and that in elt-3-deficient mutants tdc-1 expression is low (Figure 3 E).

First, only two determinations are shown in Figure 3 E (where the mRNA of various elt-3 targets is measured). I think it is too premature to draw conclusions with n=2. It is important to increase the n, especially considering that this is a relevant figure for the whole conclusion of the paper.

Assuming that the conclusions obtained in Figure 1 E are valid, the reasoning that MGH-1 induces the tyraminergic signal through the activation of the TF elt-3 sounds logical. However, this hypothesis does not agree with other data. For instance, MGH-1 treatment does not significantly modulate tdc-1 expression (Figure 1 F). Ser-2 expression is induced upon MGH-1 treatment (Figure 1F) but does not respond to elt-3 (Figure 1E). This raises questions as to how all these players interrelate to modulate the increase in feeding due to AGEs.

It may be possible that the increase in elt-3 upon MGH1 treatment is not sufficient to positively modulate tdc-1 Expression? In this context, one can think that a basal level of elt-3 is needed because otherwise, tyramine release would be low (and this explains the glod-4, tdc-1 and glod-4; elt-3 phenotypes?).

tdc-1 rescue assays with promoters lacking the elt-3 TF binding sequence could demonstrate how important the modulation of tdc-1 expression by elt-3 is during MGH-1 accumulation.

2) Results page 6 (line 117): As exogenous serotonin increases bacterial clearance in glod-4 animals, the authors claim that the increase in pharyngeal pumping in these animals is independent of serotonin. This is not concluded from this experiment. It may in fact be the case that even though it depends on endogenous serotonin release, the exogenous addition of serotonin at a high concentration may further induce pharyngeal pumping. The experiment in the manuscript that actually shows that the induction of pharyngeal pumping by MGH-1 does not depend on serotonin is shown in Figure suppl. 2 A, where MGH-1 treatment induces pumping even in tph-1 worms. To assume that the increase in pumping in glod-4 is independent of 5ht, one would have to compare the pharyngeal pumping (and bacterial clearance) of tph-1 and glod-4;tph-1 worms.

3) Results Page 7 "In addition, we also demonstrated that MG-H1 regulates pharyngeal pumping rate in a dose-dependent manner (Figure 1E)". To assert this, one would have to compare the pharyngeal pumping between the MGH-1 conditions themselves, rather than doing a student test against the naive condition. In addition, it would be advisable to do the experiments on the same day (given the variation in pharyngeal pumping in the naive condition. For example, the first naive shows a lot of dispersion and the third naive condition seems to have a lower average than the first one).

4) Results Page 7 (line 130). "In addition to food consumption, glod-4 mutant exhibited a significantly increased preference towards food source OP50-1 at day 1 and day 3 of adulthood compared to wild-type N2 worms (Figure 1F + Figure Suppl. 1F). Furthermore, we noticed that wild-type N2 worms preferred exogenous MG-H1 compared to MGO when provided with bacterial food source *E. coli* OP50-1. We did not observe this phenotype in the glod-4 mutant background (Figure 1G), suggesting that MG-H1 and glod-4 null mutation increases feeding by overlapping mechanism."

While the food race experiments may be interesting, the connection with increased food intake is, at the very least, very indirect. The experiment that would actually demonstrate that MGH1 and glod-4 null mutation increases feeding by overlapping mechanism would be the bacterial clearance assay shown in Figure 1D, where the addition of MGH1 does not increase bacterial clearance in a glod-4 mutant background. Something similar could be done with pharyngeal pumping, to confirm the overlapping mechanisms.

With respect to figure 1G, what would be the value corresponding to the attraction to MGO? It is not indicated on the axis.

Beyond that, these food race experiments should not be included in the context of enhanced feeding rate.

5) There is a very large variation in wild-type pharyngeal pumping between experiments. For example, in some experiments it is as high as (160 pp/30sec. Figure 3c) and in others it can be as low as (110 pp/30sec. Figure 2Suppl.) While I understand that for the supplementary figures the authors performed a quick count the difference is still large. Even within the main figures there is a high variation in the controls (see, for example, Figures 2 E, F and G). While I understand that experiments such as pharyngeal pumping can be variable between experiments, I am particularly struck by such variation. Are all the experiments done with animals of the same age and raised under exactly the same conditions?

6) It is not clear, and the authors do not present any model of the mechanism by which MGH1 excess modulates the tyraminergic signal to increase food intake. It is only shown that elt-3 may be important and that tdc-1, ser-2 and tyra-2 appear to be relevant. There are no rescue or functional experiments showing where these players would be playing their role.

For example, would tyramine release be affected at the level of RIM, RIC or UV cells? Perhaps it would be good to specifically silence these neurons (e.g. with the expression of histamine-triggered anion channels, see Pokala et al., PNAS, 2014 or De Rosa et al., Nature, 2019) and see if MGH-1 is able to induce pharyngeal pumping.

It is also unclear where elt-3 is relevant. It would be important to express this TF specifically in tdc-1 expressing cells and analyze whether the response is similar to wild-type.

This is one of my main concerns regarding the manuscript. The lack of mechanistic details to understand how AGEs accumulation increases food intake argues against the significance of the work.

7) Exogenous TA has been shown to reduce pharyngeal pumping (Greer et al., 2008). Furthermore, the absence of SER-2 suppresses the tyraminergic inhibition of exacerbated pharyngeal pumping in serotonin-treated worms (Rex et a, 2004. https://doi.org/10.1111/j.1471-4159.2004.02787.x).

Since the authors' results point more towards a positive modulation of pharyngeal pumping by tyramine and ser-2 , it would be important to discuss this paradox further.

8) Many of the strains listed have not been backcrossed to clean up the background. For example: RB1690 ser-2(ok2103); VC343 glod-4(gk189); VC125 tyra-3(ok325); RB745 ser-4(ok512), etc. For tyra-2 it is not indicated which strain is used.

If backcrosses have not been performed, it is essential to perform the experiments with the clean strains. If they have done them, the strain they used for the experiments is not called as they named it. Please clarify this.

At the same time, it would be advisable that in the absence of rescue experiments, they use other alleles to confirm the observed phenotypes.

*Reviewer #2 (Recommendations for the authors):*

The study is generally well done, but the strains don't appear to be outcrossed, and none of the effects are rescued. At least for the main results, tdc-1 and elt-3, there are multiple mutants available at CGC. The authors should rescue the original mutant or show the same results using a different allele. It would also be helpful for the reader to show the alleles in the figure legend.

*Reviewer #3 (Recommendations for the authors):*

There are several points that need to be substantially improved.

1. The writing of the manuscript needs to be improved substantially. The introduction is very confusing with too many abbreviations. I would suggest a diagram to show the reaction of α-dicarbonyl (α-DG), AGEs, glyoxalases. In each of these terms, you can list the examples in that category, for example: α-DG (MGO, GO, 3DG etc.), AGEs (MG-H1, CEL, CML, GOLD, MOLD), glyoxalases (glod-4, glod-1, etc.). The introduction is not organized with a focus in each paragraph. Transitions between sentences need to be improved substantially too.

2. It is not explained or addressed in the manuscript why glod-4 KO in which we would expect less MG-H1, and MG-H1 supplementation both increase pumping. It will be helpful to measure MG-H1 level in glod-4 KO. This will also help to understand the result that the pumping rate increases are MG-H1 dose-dependent, however, glod-4 KO and MG-H1 supplementation is not additive in the increase of pumping.

3. Figure 2B shows all 66 genes are upregulated in glod-4 compared to WT. A concern is whether the gene expression is properly normalized. Are there downregulated genes other than the 66 genes? How many genes are up- and down- regulated in glod-4 vs. WT. All these necessary descriptions are not mentioned in the results part. The labeling of both A and B panels are too small for recognizing the gene and pathway names. Among the 66 "neurotransmitters and feeding genes", how many of them are neurotransmitter genes and how were they defined? Are tph-1, tdc-1, tbh-1, cat-2 genes in this list? Similar question, how were the "feeding genes" defined?

4. Figure 2B used glod-4 RNAi while other results pumping data were collected on glod-4KO. Why glod-4 KO was not used in transcriptomics data. And please label on panel B indicating glod-4 RNAi. And it is described in the method part that RNA-seq was done on "N2 and mutant animals" but not RNAi.

5. tdc-1 and the two tyramine receptors knockouts abrogate pumping rate by MG-H1 and glod-4. This would indicate that tyramine release is necessary for the pumping rate increase by MG-H1 and glod-4. Did the author try rescue the abrogation by tyramine supplementation? This is a valid control to strengthen the hypothesis.

6. The authors previously showed that glod-4 KO shortens lifespan, and supplementation of glod-4 substrates MGO also shortens lifespan. The results in Figure 4 of lifespan rescue by tdc-1 is not sufficient to support that increased feeding shortens glod-4 KO lifespan suggested in the manuscript. The valid conclusion from this data is that the lifespan decreases by glod-4 KO requires tyramine synthesis and probably tyramine signaling since the two receptors are also required for the shorter lifespan. Does MG-H1 addition rescue or further shorten glod-4 KO lifespan since MG-H1 also increases pumping rate not additively to glod-4 KO? Figure 4D labeling is too small.

---

## [Author Response]

Essential revisions:1) As you will see from the reviews attached below, all reviewers found this manuscript interesting and were enthusiastic about the central idea of the work. However, all three reviewers also agreed that there were mechanistic deficiencies in the manuscript that make it unsuitable for publication in eLife without additional data. In particular the reviewers agreed that mechanistic data about how MGH1 acts on the tyraminergic pathway to potentiate food intake were lacking in several places and weak in others (e.g., the elt-3 and tdc-1 data), and require strengthening to better support the proposed model. Specific rescues (elt-3 and tdc-1), functional experiments, and manipulation of tyramine-producing cell activity are all important approaches to obtain mechanistic information that will help identify how some of these players interact with each other (e.g., the cells and circuits that govern the processes).

We thank the editors and reviewers for their time in reviewing our manuscript. We also appreciate the strengths pointed out by the reviewers and their constructive feedback. We address these concerns below.

To strongly demonstrate the necessity of tyramine signaling in *glod-4* mutants to increase pharyngeal pumping, we supplemented tyramine exogenously (instead of laser ablation/manipulation of tyramine-producing cells) to either *elt-3;glod-4* or *tdc-1;glod-4* double mutants and analyzed the pumping behavior. Further, we chose to supplement tyramine as a rescue strategy in the mutants that lack tyramine signaling. Our new results in Figure B+C demonstrate that exogenous tyramine treatment of double mutants increased their pumping levels to *glod-4* single mutant worms. Thus, supporting the hypothesis that lack of *elt-3* reduces tyramine signaling (note the reduced expression of *tdc-1* in *elt-3* mutants, Figure 4E) to suppress the pumping in *elt-3;glod-4* double mutants. This data also demonstrates the importance of *elt-3* in mediating increased pumping in *glod-4* mutants via tyramine. Similarly, exogenous tyramine supplementation in *tdc-1;glod-4* double mutant also increased pharyngeal pumping to *glod-4* single mutant levels. However, the pumping is not increased in double mutants such as *ser-2;glod-4,* and *tyra-2;glod-4* which lacks tyramine receptors (Figure D+E).

Further, we have shown in Figure 4F that MG-H1 treatment of N2 wildtype worms modulates the expression levels of the gene in the tyramine signaling pathway, i.e., increases the expression of *elt-3* as well as *ser-2* which are essential for increased feeding in *glod-4* mutants (Figure 3+4). Thus, either exogenous treatment of MG-H1 or *glod-4* mutants (with increased accumulation of MG-H1, Figure2—figure supplement 1H+1I) increased tyramine signaling by modulating genes in the tyramine pathway. The mechanism is explained in the discussion of the revised manuscript on pages 21 and 22.

However, treatment of N2 wildtype worms with tyramine resulted in a significant decrease in the pharyngeal pumping (Figure 5A). Also, tyramine is shown to suppress pharyngeal pumping in N2 wildtype worms (1). These findings demonstrate that tyramine increases pharyngeal pumping only in the *glod-4* mutant background but decreases pumping in the N2 wildtype background. The mechanism behind this behavioral switch upon activation of the tyramine signaling pathway (i.e., from a suppressor of pharyngeal pumping in wildtype into a stimulator in *glod-4* mutant background) remains elusive. Although *tdc-*1 is expressed only in RIC, RIM, UV2 cells and gonadal sheath cells, tyramine receptors are located at distinct tissues (1) (2); thus, it can be speculated that AGEs, especially MG-H1, can modify the tissue-specific expression of tyramine receptors resulting in observed behavioral switch in response to tyramine signaling. This hypothesis can only be addressed by methodologies such as single-cell RNA sequencing which is beyond the scope of the current study.

2) Additional experiments, such as bacterial clearance (reviewer 1, comment 4), and something to rectify reviewer 3, comment 2, should also be completed.

We have addressed reviewer 1-comment 4 and reviewer 3-comment 2 with appropriate experiments and explanation, please refer below.

3) Additionally, please note that both reviewer 1 and reviewer 3 point out several instances in the writing and/or figures that require clarification/revision, although these revisions may or may not require data depending on the response (e.g., pumping controls, backcrossing of worms). All reviewer comments are included below for individual responses.

All the comments that do not require additional experiments but require appropriate explanation are revised as suggested by the reviewers in the new revised manuscript. Please see the rebuttal below.

Reviewer #1 (Recommendations for the authors):These are the points that I feel need to be addressed:1) The authors claim that MGH-1 modulates (positively) the tyraminergic signal through the activation of the TF elt-3. This is based on the fact that mutations in both tdc-1 and elt-3 suppress the exacerbated pharyngeal pumping of glod-4 mutants (Figure 2E and 3D), that elt-3 expression is increased in MGH1-treated worms (Figure 3F) and that in elt-3-deficient mutants tdc-1 expression is low (Figure 3 E).First, only two determinations are shown in Figure 3 E (where the mRNA of various elt-3 targets is measured). I think it is too premature to draw conclusions with n=2. It is important to increase the n, especially considering that this is a relevant figure for the whole conclusion of the paper.

As recommended by the reviewer, we have now increased the sample size to 5 biological replicates (Figure 4E). Our data shows that *tdc-1* expression levels are significantly reduced in *elt-3* mutant worms. Further, we now have data demonstrating that exogenous tyramine treatment on *elt-3;glod-4* double mutant increased the pharyngeal pumping similar to *glod-4* single mutant worms (Figure 5B) indicating that *elt-3* is necessary for tyramine signaling.

However, it can be observed that *ser-2* expression significantly increased in *elt-3* mutant worms which can be inferred as a compensatory mechanism for lack of tyramine signaling by increasing the expression of tyramine receptor.

Assuming that the conclusions obtained in Figure 1 E are valid, the reasoning that MGH-1 induces the tyraminergic signal through the activation of the TF elt-3 sounds logical. However, this hypothesis does not agree with other data. For instance, MGH-1 treatment does not significantly modulate tdc-1 expression (Figure 1 F). Ser-2 expression is induced upon MGH-1 treatment (Figure 1F) but does not respond to elt-3 (Figure 1E). This raises questions as to how all these players interrelate to modulate the increase in feeding due to AGEs.It may be possible that the increase in elt-3 upon MGH1 treatment is not sufficient to positively modulate tdc-1 Expression?. In this context, one can think that a basal level of elt-3 is needed because otherwise, tyramine release would be low (and this explains the glod-4, tdc-1 and glod-4; elt-3 phenotypes?).tdc-1 rescue assays with promoters lacking the elt-3 TF binding sequence could demonstrate how important the modulation of tdc-1 expression by elt-3 is during MGH-1 accumulation.

We show that MG-H1 treatment modulates factors in the *elt-3-tdc-1*/tyramine pathways (Figure 4F, especially increased expression of *elt-3* and *ser-2* genes) to increase pharyngeal pumping. In order to strengthen the involvement of tyramine signaling, we supplemented tyramine (tyramine rescue) to either *elt-3;glod-4,* or *tdc-1;glod-4* double mutant instead of *tdc-1* rescue suggested by the reviewer. Our results in Figure B+C demonstrate that exogenous tyramine treatment of double mutants increased the pumping levels to *glod-4* single mutant worms. Thus, we can understand that lack of *elt-3* modulates tyramine levels to suppress the pumping in *elt-3;glod-4* double mutants. Similarly, exogenous tyramine supplementation in *tdc-1;glod-4* double mutant also increased pharyngeal pumping to *glod-4* single mutant levels. However, the pumping is not increased in double mutants such as *ser-2;glod-4,* and *tyra-2;glod-4* which lack tyramine receptors.

Further, from the literature we can find that tyramine suppresses the pharyngeal pumping in *C. elegans* (1). Thus, treatment of N2 wildtype worms with tyramine resulted in a significant decrease in the pharyngeal pumping (Figure 5A). These findings demonstrate that tyramine increases pharyngeal pumping only in *glod-4* mutant background but decreases pumping in N2 wildtype background. And further analysis is required to understand the mechanistic shift in the tyramine signaling, however it is beyond the scope of this research article.

2) Results page 6 (line 117): As exogenous serotonin increases bacterial clearance in glod-4 animals, the authors claim that the increase in pharyngeal pumping in these animals is independent of serotonin. This is not concluded from this experiment. It may in fact be the case that even though it depends on endogenous serotonin release, the exogenous addition of serotonin at a high concentration may further induce pharyngeal pumping. The experiment in the manuscript that actually shows that the induction of pharyngeal pumping by MGH-1 does not depend on serotonin is shown in Figure suppl. 2 A, where MGH-1 treatment induces pumping even in tph-1 worms. To assume that the increase in pumping in glod-4 is independent of 5ht, one would have to compare the pharyngeal pumping (and bacterial clearance) of tph-1 and glod-4;tph-1 worms.

As recommended by the reviewer, we compared the pharyngeal pumping in *tph-1 (mg280)* and *tph-1;glod-4* mutants background (Figure 2—figure supplement 1D). Worms lacking *tph-1* (tryptophan hydroxylase) enzyme, as well as *tph-1;glod-4* double mutants, which lack serotonin (1) (3) show decreased pumping compared to N2 wildtype worms; however, *tph-1;glod-4* double mutant worms show significantly increased pumping compared to *tph-1* single mutants demonstrating that *glod-4* mutation dependent increase in pharyngeal pumping is independent of serotonin. Also, note that MG-H1 treatment significantly increased pharyngeal pumping in *tph-1* mutants in Figure 3—figure supplement 2A which supports our claim that *glod-4* mutant induced pumping is independent of serotonin.

3) Results Page 7 "In addition, we also demonstrated that MG-H1 regulates pharyngeal pumping rate in a dose-dependent manner (Figure 1E)". To assert this, one would have to compare the pharyngeal pumping between the MGH-1 conditions themselves, rather than doing a student test against the naive condition. In addition, it would be advisable to do the experiments on the same day (given the variation in pharyngeal pumping in the naive condition. For example, the first naive shows a lot of dispersion and the third naive condition seems to have a lower average than the first one).

Our conclusion is based on the stronger significance (lower p value) caused by reduced dispersion of the data at higher concentration of MG-H1. This indicates that higher concentrations of MG-H1 can increase the pharyngeal pumping in almost all the treatment worms thus predisposing the worms with lower pumping rates in the Gaussian distribution to higher pumping levels. Now, it is made clear in the manuscript on page 18.

It is important to explain that there were a few technical difficulties in recording pumping videos from several treatment groups in a narrow window of time. From Figure 2A it is very evident that pharyngeal pumping is very dynamic which changes with the worm’s development. Further, it takes about a minute to record a 40 sec pumping video which includes finding a worm and focusing to start the recording. Approximately 26 to 30 worms were recorded per treatment group which requires 1 hour (including personal fatigue experienced by the recorder). Dynamic pumping behavior can induce variations between the worms recorded in the first treatment group and the last treatment group especially when there are 5 treatment groups to record pumping videos which takes around 5 hours. Thus, we decided to record the minimum required treatment groups as possible to reduce variations caused by delayed time and worms’ development. In case, when more treatment/genetic groups need to be compared the video recording between different groups are staggered (recording of 10 worms per group followed by recording worms from other groups and repeating this cycle until 30 worms per group were recorded) to minimize the various caused.

Further, variations are also induced by fluctuations in the lab temperature, which is overcome by preparing 3 plates of worms for a single treatment group and only 10 worms are recorded per plate while the other plates are incubated at 20 °C.

An insight into the above-mentioned strategies has been updated on the “Materials and methods” section on page 25.

4) Results Page 7 (line 130). "In addition to food consumption, glod-4 mutant exhibited a significantly increased preference towards food source OP50-1 at day 1 and day 3 of adulthood compared to wild-type N2 worms (Figure 1F + Figure Suppl. 1F). Furthermore, we noticed that wild-type N2 worms preferred exogenous MG-H1 compared to MGO when provided with bacterial food source *E. coli* OP50-1. We did not observe this phenotype in the glod-4 mutant background (Figure 1G), suggesting that MG-H1 and glod-4 null mutation increases feeding by overlapping mechanism."While the food race experiments may be interesting, the connection with increased food intake is, at the very least, very indirect. The experiment that would actually demonstrate that MGH1 and glod-4 null mutation increases feeding by overlapping mechanism would be the bacterial clearance assay shown in Figure 1D, where the addition of MGH1 does not increase bacterial clearance in a glod-4 mutant background. Something similar could be done with pharyngeal pumping, to confirm the overlapping mechanisms.With respect to figure 1G, what would be the value corresponding to the attraction to MGO? It is not indicated on the axis.Beyond that, these food race experiments should not be included in the context of enhanced feeding rate.

As recommended by the reviewer, we performed pharyngeal pumping quantification on *glod-4* mutant worms treated with 150 µM MG-H1 and compared with N2 and *glod-4* either treated with arginine or MG-H1 (Author response image 1). Our data demonstrates that *glod-4* ARG does not differ significantly with *glod-4* MG-H1 suggesting both *glod-4* mutants and MG-H1 treatment modulates similar pathway to increase feeding. From Figure 4F, it is also evident that MG-H1 treatment induces gene expression changes in *elt-3*-tyramine signaling pathways which also validates that MG-H1 acts through *elt-3*-tyramine signaling pathway.

**Author response image 1. sa2fig1:** Quantification of pharyngeal pumping. One way ANOVA with Fisher’s LSD test. **** p<0.0001. Error bar ± SD.

We thank the reviewer for pointing out the lack of description in old Figure 1G. We apologize that a small proofreading error resulted in a wrong graph which is updated now with two graphs as shown in Figure 2G and 2H. Figure 2G is a comparison between N2 wildtype worm treated with either MGO or MG-H1 with untreated control, showing that mixing of MGO to OP50-1 did not significantly increase the attraction of worms towards food, unlike MG-H1. Figure 2H demonstrates that *glod-4* mutant’s attraction towards food is unaffected by MG-H1 when mixed with OP50-1.

5) There is a very large variation in wild-type pharyngeal pumping between experiments. For example, in some experiments it is as high as (160 pp/30sec. Figure 3c) and in others it can be as low as (110 pp/30sec. Figure 2Suppl.) While I understand that for the supplementary figures the authors performed a quick count the difference is still large. Even within the main figures there is a high variation in the controls (see, for example, Figures 2 E, F and G). While I understand that experiments such as pharyngeal pumping can be variable between experiments, I am particularly struck by such variation. Are all the experiments done with animals of the same age and raised under exactly the same conditions?

Yes, the worms were grown in the same conditions and the videos were recorded at similar ages, ± few minutes difference, throughout the project. As pointed out by the reviewer pharyngeal pumping is variable and dynamic. Thus, we realized the importance of validating our findings with alternative methodologies, for example – very important pumping data was supported by food clearance assay (Figure 2B+2D in the manuscript). Further, a few important pharyngeal pumping counts were performed double-blinded to minimize the bias.

It is important to explain that there were a few technical difficulties in recording pumping videos from several treatment groups in a narrow window of time. From Figure 2A it is very evident that pharyngeal pumping is very dynamic which changes with the worm’s development. Further, it takes about a minute to record a 40 sec pumping video which includes finding a worm and focusing to start the recording. Approximately 26 to 30 worms were recorded per treatment group per biological replicate which requires 1 hour (including personal fatigue experienced by the recorder). Dynamic pumping behavior can induce variations between the worms recorded in the first treatment group and the last treatment group especially when there are 5 treatment groups to record pumping videos which takes around 5 hours. Thus, we decided to record the minimum required treatment groups as possible to reduce variations caused by delayed time and worms’ development. In case, when more treatment/genetic groups need to be compared the video recordings between different groups are staggered (recording of 10 worms per group followed by recording worms from other groups and repeating this cycle until 30 worms per group were recorded) to minimize the variations.

Further, variations are also induced by fluctuations in the lab temperature, which is overcome by preparing 3 plates of worms for a single treatment group and only 10 worms are recorded per plate while the other plates are incubated at 20 °C.

To further average out the variations, the biological repeats were pooled together to represent the final data which is already explained in the “Materials and methods” section on Page 25.

Also, some odd-behaving worms were not recorded such as worms that are stationary for prolonged amounts of time, worms that did not show any pharyngeal pumping, and worms that are potentially injured with any visible damages, etc.

An insight on the above-mentioned strategies has been updated on the “Materials and methods” section on page 25.

6) It is not clear, and the authors do not present any model of the mechanism by which MGH1 excess modulates the tyraminergic signal to increase food intake. It is only shown that elt-3 may be important and that tdc-1, ser-2 and tyra-2 appear to be relevant. There are no rescue or functional experiments showing where these players would be playing their role.For example, would tyramine release be affected at the level of RIM, RIC or UV cells? Perhaps it would be good to specifically silence these neurons (e.g. with the expression of histamine-triggered anion channels, see Pokala et al., PNAS, 2014 or De Rosa et al., Nature, 2019) and see if MGH-1 is able to induce pharyngeal pumping.It is also unclear where elt-3 is relevant. It would be important to express this TF specifically in tdc-1 expressing cells and analyze whether the response is similar to wild-type.This is one of my main concerns regarding the manuscript. The lack of mechanistic details to understand how AGEs accumulation increases food intake argues against the significance of the work.

To further demonstrate the role of tyramine signaling in mediating MG-H1-induced increased pharyngeal pumping, as well as in *glod-4* mutants, we supplemented tyramine to either *elt-3;glod-4* or *tdc-1;glod-4* double mutant and analyzed the pumping behavior. Our results in Figure 5B-E demonstrate that exogenous tyramine treatment of double mutants increased the pumping levels to *glod-4* single mutant worms. Thus, we show that lack of *elt-3* modulates tyramine signaling to suppress the pumping in *elt-3;glod-4* double mutants. This demonstrates the importance of *elt-3* in mediating increased pumping in *glod-4* mutants via tyramine. Similarly, exogenous tyramine supplementation in *tdc-1;glod-4* double mutant also increased pharyngeal pumping to *glod-4* single mutant levels. However, the pumping is not increased in double mutants such as *ser-2;glod-4,* and *tyra-2;glod-4* which lacks tyramine receptors.

Silencing the tyramine synthesis neurons to dissect the neuronal circuitry is an interesting experiment, as suggested by the reviewers; however, it is very time-consuming and beyond the scope of this study. We have demonstrated that lack of either tyramine or tyramine receptor suppressed the pumping in the *glod-4* mutant background which was rescued by exogenous tyramine in *elt-3;glod-4* and *tdc-1;glod-4* double mutant.

7) Exogenous TA has been shown to reduce pharyngeal pumping (Greer et al., 2008). Furthermore, the absence of SER-2 suppresses the tyraminergic inhibition of exacerbated pharyngeal pumping in serotonin-treated worms (Rex et a, 2004. https://doi.org/10.1111/j.1471-4159.2004.02787.x).Since the authors' results point more towards a positive modulation of pharyngeal pumping by tyramine and ser-2 , it would be important to discuss this paradox further.

Treatment of N2 wildtype worms with tyramine resulted in a significant decrease in the pharyngeal pumping (Figure 5). Tyramine suppressing pharyngeal pumping has already been described in the literature (1), as well as indicated by the reviewer. However, our data from *elt-3;glod-4* and *tdc-1;glod-4* double mutants show that tyramine treatment increased the pumping to *glod-4* single mutant levels (Figure B+C). Also, double mutants lacking tyramine receptors were not rescued by tyramine (Figure D+E). These findings demonstrate that tyramine increases pharyngeal pumping only in *glod-4* mutant background but decreases pumping in N2 wildtype background.

The mechanism behind this behavioral switch upon activation of the tyramine signaling pathway (i.e., from a suppressor of pharyngeal pumping in wildtype into a stimulator in *glod-4* mutant background) remains elusive. Although *tdc-1* is expressed only in RIC, RIM, UV2 pair of cells and gonadal sheath cells, tyramine receptors are located at distinct tissues (1) (2); thus, it can be speculated that AGEs, especially MG-H1, can modify the tissue-specific expression of tyramine receptors resulting in observed behavioral switch in response to tyramine signaling. This hypothesis can only be addressed by methodologies such as single-cell RNA sequencing which is beyond the scope of the current study. Currenty, we discussed this paradox in the ‘Discussion’ section on pages 21 and 22.

8) Many of the strains listed have not been backcrossed to clean up the background. For example: RB1690 ser-2(ok2103); VC343 glod-4(gk189); VC125 tyra-3(ok325); RB745 ser-4(ok512), etc. For tyra-2 it is not indicated which strain is used.If backcrosses have not been performed, it is essential to perform the experiments with the clean strains. If they have done them, the strain they used for the experiments is not called as they named it. Please clarify this.At the same time, it would be advisable that in the absence of rescue experiments, they use other alleles to confirm the observed phenotypes.

All the strains are outcrossed at least 3 times or more with N2 wildtype strain obtained from CGC, we have updated the “Materials and methods” now.The strain used for *tyra-2* is now mentioned in the “Materials and methods” as FX1846 *tyra-2 (tm1846)*.As recommended by the reviewer, we have used another allelic mutant of *tdc-1 (n3420)* and evaluated its pumping in *n3420* as well as in *tdc-1(n3420);glod-4* double mutant (Author response image 2). Our results demonstrate that *glod-4* single mutant shows significantly increased pumping compared to N2 wildtype worms; however, *tdc-1 (n3420);glod-4 (gk189)* double mutant worms do not show any significant difference indicating *tdc-1* enzyme is essential for increased feeding in *glod-4* single mutants.Similarly, we also evaluated the role of *elt-3* in mediating the pharyngeal pumping in *glod-4* single mutants by knockdown *elt-3* using RNAi (Author response image 2). Again, our results demonstrate that *elt-3* is essential for increased feeding in *glod-4* single mutants.

**Author response image 2. sa2fig2:** Quantification of pharyngeal pumping in N2 wildtype, *glod-4 (gk189)*, *tdc-1 (n3420)* as well as in double mutants (A) and with *elt-3* knockdown (B). One way ANOVA with Fisher’s LSD test. **** p<0.0001. Error bar ± SD.

Reviewer #2 (Recommendations for the authors):The study is generally well done, but the strains don't appear to be outcrossed, and none of the effects are rescued. At least for the main results, tdc-1 and elt-3, there are multiple mutants available at CGC. The authors should rescue the original mutant or show the same results using a different allele. It would also be helpful for the reader to show the alleles in the figure legend.

We thank the reviewer for pointing out the importance of using either multiple allelic mutants or rescue of the mutant phenotype to validate, as well as strengthen, the role of *tdc-1* and *elt-3* genes in modulating the feeding behavior.

The strains used in this study are outcrossed at least 3 times, which is now indicated in the ‘Materials and methods’ section on Page 24.As requested by the reviewer, we now included pharyngeal pumping data from another allelic mutant of MT10661 *tdc-1*(n3420) to demonstrate that lack of *tdc-1* suppresses pharyngeal pumping in *glod-4;tdc-1* double mutant. Also, we knocked down the *elt-3* in *glod-4* mutants worms using RNAi feeding approach and demonstrated that *elt-3* is essential of increased feeding in *glod-4* mutants (Author response image 2) .Finally, we rescued increased pharyngeal pumping (glyoxalase *glod-4* mutant phenotype) in either *tdc-1;glod-4* or *elt-3;glod-4* double mutants with exogenous administration of tyramine. This result shows that tyramine signaling is essential for increased feeding in glyoxalase *glod-4* mutants which is suppressed with the lack of either *tdc-1* or *elt-3* genes. However, exogenous tyramine did not increase pharyngeal pumping in tyramine receptor-*glod-4* double mutants.

Reviewer #3 (Recommendations for the authors):There are several points that need to be substantially improved.1. The writing of the manuscript needs to be improved substantially. The introduction is very confusing with too many abbreviations. I would suggest a diagram to show the reaction of α-dicarbonyl (α-DG), AGEs, glyoxalases. In each of these terms, you can list the examples in that category, for example: α-DG (MGO, GO, 3DG etc.), AGEs (MG-H1, CEL, CML, GOLD, MOLD), glyoxalases (glod-4, glod-1, etc.). The introduction is not organized with a focus in each paragraph. Transitions between sentences need to be improved substantially too.

We thank the reviewer for pointing out an unclear Introduction. As per reviewer’s request, we now introduced a pictorial diagram in the Figure 1 to explain the interactions between dicarbonyls, AGEs and glyoxalases enzymes as well as explained it in the pages 3 and 4. Also, a detailed understanding about dicarbonyls, AGEs, glyoxalases, and their impact on the health as well as disease can be obtained from a review from our lab, Chaudhuri *et la.* 2018 (4).

2. It is not explained or addressed in the manuscript why glod-4 KO in which we would expect less MG-H1, and MG-H1 supplementation both increase pumping. It will be helpful to measure MG-H1 level in glod-4 KO. This will also help to understand the result that the pumping rate increases are MG-H1 dose-dependent, however, glod-4 KO and MG-H1 supplementation is not additive in the increase of pumping.

It has been previously shown that *glod-4* KO worms show more MGO (5). We now further clarify that glyoxalase enzymes detoxify dicarbonyls (such as MGO, GO, etc.) and the lack of glyoxalase enzymes results in accumulation of dicarbonyls which interacts with biomolecules to increase the formation and accumulation of AGEs in pages 3 and 4 (Figure 1). Thus, *glod-4* glyoxalase mutant worms accumulate more MG-H1 which signals to increase the feeding which is comparable to increased feeding after exogenous treatment of MG-H1. Now, we also quantified the levels of MG-H1 in *glod-4* mutant worms using mass spectrometry (LC-MRM) and included the data in Figure 2—figure supplement 1H+1I, which shows significantly increased MG-H1 levels.

3. Figure 2B shows all 66 genes are upregulated in glod-4 compared to WT. A concern is whether the gene expression is properly normalized. Are there downregulated genes other than the 66 genes? How many genes are up- and down- regulated in glod-4 vs. WT. All these necessary descriptions are not mentioned in the results part. The labeling of both A and B panels are too small for recognizing the gene and pathway names. Among the 66 "neurotransmitters and feeding genes", how many of them are neurotransmitter genes and how were they defined? Are tph-1, tdc-1, tbh-1, cat-2 genes in this list? Similar question, how were the "feeding genes" defined?

The RNAseq is normalized by FPKM (fragments per kilobase of exon per million fragments mapped) method.The total number of genes identified in the RNAseq analysis is 20,277. The total genes that changed significantly are 5035 with upregulated genes amounting to 2237 and downregulated amounts to 2798 genes. This information is now included on page 10 of the manuscript.Gene ontology was performed in DAVID v6.8 which identified neurotransmitter and feeding genes which is mentioned in the “Materials and methods” section. Because neurotransmitters also regulate feeding behavior, it is essential to understand that a clear separation between neurotransmitter genes and feeding genes is difficult.Gene *tph-1* was identified in RNAseq but there was no significant difference between N2 and *glod-4* RNAi. However, RNAseq did not identify the following genes *tdc-1*, *tbh-1,* and *cat-2*.

4. Figure 2B used glod-4 RNAi while other results pumping data were collected on glod-4KO. Why glod-4 KO was not used in transcriptomics data. And please label on panel B indicating glod-4 RNAi. And it is described in the method part that RNA-seq was done on "N2 and mutant animals" but not RNAi.

As requested by the reviewer, we labeled the RNAseq heatmap appropriately (Figure 3A). We prepared samples for RNAseq from both *glod-4* KO worms and by *glod-4* RNAi knockdown. During further processing, samples from *glod-4* RNAi worms demonstrated the best quality for RNAseq analysis. However, samples from *glod-4* KO required further standardization to pass the quality control. Thus, we utilized samples from *glod-4* RNAi instead of samples from *glod-4* null mutants for RNAseq analysis.

5. tdc-1 and the two tyramine receptors knockouts abrogate pumping rate by MG-H1 and glod-4. This would indicate that tyramine release is necessary for the pumping rate increase by MG-H1 and glod-4. Did the author try rescue the abrogation by tyramine supplementation? This is a valid control to strengthen the hypothesis.

As recommended by the reviewer, we exogenously supplemented tyramine in different genetic backgrounds and demonstrated its role in modulating pharyngeal pumping. Tyramine supplementation in N2(wt) worms significantly reduced the pumping at various concentrations (Figure 5A). However, tyramine supplementation in double mutants such as *elt-3;glod-4* and *tdc-1;glod-4* resulted in a significant increase in the pumping comparable to *glod-4* single mutants. Further, tyramine supplementation in tyramine receptor mutants leads to no changes in the pumping. This result demonstrates that in wildtype background tyramine suppresses pumping consistent with the existing literature (1); however, in the *glod-4* mutant background tyramine increases pumping. Thus far, the mechanisms by which tyramine increased the pumping (signaling switch from a suppressor in wildtype worms into pharyngeal pumping activator in *glod-4* mutant background) in *glod-4* mutants remains elusive and needs further studies in the future.

6. The authors previously showed that glod-4 KO shortens lifespan, and supplementation of glod-4 substrates MGO also shortens lifespan. The results in Figure 4 of lifespan rescue by tdc-1 is not sufficient to support that increased feeding shortens glod-4 KO lifespan suggested in the manuscript. The valid conclusion from this data is that the lifespan decreases by glod-4 KO requires tyramine synthesis and probably tyramine signaling since the two receptors are also required for the shorter lifespan. Does MG-H1 addition rescue or further shorten glod-4 KO lifespan since MG-H1 also increases pumping rate not additively to glod-4 KO? Figure 4D labeling is too small.

We thank the reviewer for suggesting appropriate inference of the data. The conclusion was updated in the manuscript on pages 16 and 17, as suggested by the reviewer. Further, we also performed lifespan assay in N2 and *glod-4* mutant worms with 150 µM MG-H1 treatment (Figure 6—figure supplement 1). MG-H1 treatment significantly reduced the lifespan in N2 wildtype worms. Glyoxalase mutant worms showed significantly shorter lifespans than N2 worms; however, MG-H1 treatment did not further affect the lifespan of *glod-4* mutant worms.We enlarged figure 6D (old figure 4D).

References:

1. Dallière N, Holden-Dye L, Dillon J, O'Connor V, Walker RJ. *Caenorhabditis elegans* Feeding Behaviors. Oxford Research Encyclopedia of Neuroscience2017.

2. Rex E, Hapiak V, Hobson R, Smith K, Xiao H, Komuniecki R. TYRA-2 (F01E11.5): a *Caenorhabditis elegans* tyramine receptor expressed in the MC and NSM pharyngeal neurons. J Neurochem. 2005;94(1):181-91.

3. Ji Ying Sze MV, Curtis Loer, Yang Shi & Gary Ruvkun. Food and metabolic signalling defects in a *Caenorhabditis elegans* serotonin-synthesis mutant. Nature. 2000;403:560-4.

4. Chaudhuri J, Bains Y, Guha S, Kahn A, Hall D, Bose N, et al. The Role of Advanced Glycation End Products in Aging and Metabolic Diseases: Bridging Association and Causality. Cell Metab. 2018;28(3):337-52.

5. Chaudhuri J, Bose N, Gong J, Hall D, Rifkind A, Bhaumik D, et al. A *Caenorhabditis elegans* Model Elucidates a Conserved Role for TRPA1-Nrf Signaling in Reactive α-Dicarbonyl Detoxification. Curr Biol. 2016;26(22):3014-25.